# Tonal and syllabic encoding in overt Cantonese Chinese speech production: An ERP study

**Andus Wing-Kuen Wong**[1], **Ho-Ching Chiu**[1]*, **Yiu-Kei Tsang**[2], **Hsuan-Chih Chen**[3]

1 Department of Social and Behavioural Sciences, Nam Shan Psychology Laboratory, City University of Hong Kong, Hong Kong S.A.R., Hong Kong, China, 2 Department of Education Studies, Baptist University of Hong Kong, Hong Kong S.A.R., Hong Kong, China, 3 Department of Psychology, Chinese University of Hong Kong, Hong Kong S.A.R., Hong Kong, China

* silas.chiu.jw@gmail.com

**Data Availability Statement:** The minimal data set underlying the results described in this work can be found at https://zenodo.org/record/7750292 (DOI: 10.5281/zenodo.7750292).

## Abstract

This study was conducted to investigate how syllables and lexical tones are processed in Cantonese speech production using the picture-word interference task with concurrent recording of event-related brain potentials (ERPs). Cantonese-speaking participants were asked to name aloud individually presented pictures and ignore an accompanying auditory word distractor. The target and distractor either shared the same word-initial syllable with the same tone (Tonal-Syllable related), the same word-initial syllable without the same tone (Atonal-Syllable related), the same tone only (Tone alone related), or were phonologically unrelated. Participants' naming responses were faster, relative to an unrelated control, when the target and distractor shared the same tonal- or atonal-syllable but null effect was found in the Tone alone related condition. The mean ERP amplitudes (per each 100-ms time window) were subjected to stimulus-locked (i.e., time-locked to stimulus onset) and response-locked (i.e., time-locked to response onset) analyses. Significant differences between related and unrelated ERP waves were similarly observed in both Tonal-Syllable related and Atonal-Syllable related conditions in the time window of 400–500 ms post-stimulus. However, distinct ERP effects were observed in these two phonological conditions within the 500-ms pre-response period. In addition, null effects were found in the Tone alone related condition in both stimulus-locked and response-locked analyses. These results suggest that in Cantonese spoken word production, the atonal syllable of the target is retrieved first and then associated with the target lexical tone, consistent with the view that tone has an important role to play at a late stage of phonological encoding in tonal language production.

## 1. Introduction

The phonological form of an utterance needs to be retrieved and used during speech production. The cognitive processes underlying phonological form preparation have been termed

**Funding:** The work described in this paper was fully/partially supported by research grants from the Research Grants Council of the Hong Kong Special Administrative Region, China (Project No. CityU 21402514 and CityU 11673316). The funders had no role in study design, data collection and analysis, decision to publish, or preparation of the manuscript.

**Competing interests:** The authors have declared that no competing interests exist.

phonological planning. Much research has been reported on how different types of phonological units are processed in conjunction with each other during phonological planning [1–6].

Most theories of phonological planning, however, have primarily been developed for stress languages such as Dutch and English, and are usually based on speech-error or behavioural (e.g., speaker's response latency) data. In stress languages such as Dutch and English, supra-segmental information such as the stress is rarely used in distinguishing lexical items [7]. In contrast, supra-segmental information such as the tone (i.e., the level and contour of fundamental frequency realized on segmental content) is crucial for lexical distinction in tonal languages such as Cantonese or Mandarin Chinese [8,9]. Therefore, it remains open to what extent that the existing prominent theories of language production can be generalized to non-stress tonal languages, and how tone is processed together with segmental information in tonal language production.

Existing evidence regarding the role of tone in tonal language production comes from two major sources, namely speech error studies and chronometric studies. The evidence from speech error studies is mixed [10], for instance, found that the tonal errors exhibited by a group of left-hemisphere damaged Mandarin aphasic speakers were qualitatively and quantitatively similar to the segmental errors that these speakers exhibited. Likewise, in a study on slips of the tongue, [11] observed that the tonal errors made by Thai (a tone language) speakers were similar to their consonant and vowel errors. These results indicate that tones are processed or represented similarly as phonemic segments (see also [12], for consistent findings in Mandarin). Similarly, [13] investigated a large corpus of naturalistic Cantonese speech errors and found non-negligible tonal errors which behaved similarly with the segmental errors and were prone to the influence from immediate contexts. Based on the finding that tonal errors were affected by the surrounding context (e.g., lexical level processing) similar to that of the segmental errors, [13] hence concluded that tone and segments are concurrently selected at an early stage of phonological planning. This early selection and encoding of the lexical tone has been referred to as the *early encoding view* [13]. In contrast, however, [14] reported contradictory findings in which independent acoustic patterns of tonal and segmental errors were found in the case of a female Mandarin aphasic speaker. Furthermore, by analyzing speech errors made by unimpaired Mandarin speakers, [15] found distinctive patterns of tonal and segmental errors (where the genuine tonal errors were rare and much less than the segmental errors observed), suggesting that tones are processed and represented differently from segments. [15] concluded that the processing of tone might involve a mechanism which is different from the processing of segments, similar to the processing of stress in stress languages. Furthermore, [16] found higher-than-chance level of syllable (as a chunk and without the tone specified) movement errors independent of the tone in Mandarin speech, consistent with the notion that the segmental content is processed differently from the tone.

Apart from analyzing speech error patterns, researchers have adopted the experimental approach to address the issue. For instance, [17] employed the form preparation task to investigate the phonological planning processes in Mandarin di-syllabic word production. A significant facilitation effect on participants' naming latencies was found when the response words in a block shared the same word-initial syllable (with or without the same tone) but null effects were observed when the response words shared only the same onset consonant or the same tone (see [18], for convergent findings in Cantonese Chinese). [17] therefore argued that atonal syllables (i.e., syllable units with the segmental contents specified but not the tone) have their own representations and are retrieved independently from the tone at the beginning of phonological planning (consistent with [16]). The atonal syllables then associate with the tone in a later stage. [19] replicated these findings and used the term "proximate unit" to refer to the first retrievable unit in phonological planning. According to these researchers, the nature

of the proximate unit varies across languages, with segments as the proximate units in Dutch and English but atonal syllables in Mandarin [20]. Unlike the segmental contents, [17] argued that the lexical tone is mapped independently onto a structural frame (i.e., metrical frame) which specifies the tonal property of the target word. And the tone is associated with the segmental content only at a later stage of phonological encoding. As tonal encoding occurs late during phonological encoding, this notion has been referred to as the *late encoding view* [13].

In addition, [21–23] investigated the role of tone in Cantonese spoken word production using the picture-word interference (PWI) task. The PWI task is a frequently used paradigm in the language production literature and has been shown to be sensitive to the phonological planning processes (e.g., [24–27]). In a PWI task, participants are asked to name aloud individually presented pictures of concrete objects and ignore an accompanying word distractor. In a series of Cantonese PWI experiments, [21–23] found that participants' picture naming responses were faster, relative to the unrelated control, when the target and distractor shared the same syllable with or without the same tone but null effects were observed when they shared the same tone only. Furthermore, the priming effect observed in the tonal syllable related (i.e., syllable + tone related) condition was significantly larger than that in the atonal syllable related condition. The null effect of tone alone, together with the contrast between tonal and atonal syllables suggested that the effect of tone is dependent on syllabic relatedness. Similarly, [23] observed significant priming effects when the target and distractor shared two similar phonemic segments, and a significantly larger priming effect when the target and distractor shared also the same tone. These results indicated that the effect of tone alone is weak (at least not being shown in any behavioural data) and can possibly be detected upon simultaneous segmental priming.

The above reviewed speech error and behavioural data both suggested that the lexical tone has a role to play in tonal language production and is implicated in phonological planning. Although the effect of tone alone was rarely observed in the past experimental research, it's effect on speech production has been shown upon concurrent segmental priming, indicating that certain planning processes must exist which enable the joint effect of segmental and tonal priming. Accordingly, most theories of speech production for Chinese languages assume a mechanism for tonal encoding. For instances, the form encoding model by [19] assumes that in Mandarin phonological encoding, the atonal syllable of the target is retrieved first after lexical selection, a word-shape frame is separately activated which specifies the tone and syllable internal structure. Syllabified phonemic segments are then associated with the word-shape frame at a later stage. Similarly, in a modified version of the WEAVER++ (Word-form Encoding And VERification) model for Mandarin speech production, [20] proposed that the atonal syllable of the target is retrieved at the beginning of phonological encoding and the tone is mapped onto a tonal frame which specifies the tonal features of the target. It was then followed by segmental spell-out and then segment-to-tonal-frame association. The resultant phonological word with both segmental and tonal features specified determines the syllable motor programs involved in initiating and guiding the subsequent articulation processes. In short, these theories of Chinese speech production assume that tone (as represented by a structural frame) is processed separately from the retrieval of the atonal syllable which occurs at the beginning of phonological encoding. And the tone associates with the syllable constituent segments only at a later stage of phonological encoding, consistent with the late encoding view.

However, clear empirical evidence showing the detailed time course of tonal encoding and how tone is processed in conjunction with segmental information is still scarce. Although speech error data tell us a great deal about the psychological reality of tone in speech planning, there are limitations in the conclusions one can make with error data regarding the temporal dynamics of tonal and syllabic encoding in normal speech production [28]. Furthermore,

previous experimental work on tonal language production showed that the effect of tone alone is weak at best which was difficult to be detected [17,21]. More evidence is therefore needed to understand the role of tone in tonal language production and the time course of tonal encoding relative to segmental encoding. In addition, although there is evidence to suggest a joint effect of tonal and segmental priming (e.g., tonal syllable priming > atonal syllable priming), the locus of this joint effect and the relative time course of how tonal and segmental information is processed in phonological planning are still largely unknown. The present study was therefore conducted to address these issues using a more sensitive technique. For instance, researchers investigating phonological planning have recently begun to record temporal brain activity in the form of event-related brain potentials (ERPs) during overt naming tasks [29,30]. This approach has been shown to be promising in revealing the fine-grained temporal processes prior to the overt articulatory response, which are difficult to identify using traditional approaches to the study of language production [31–37]. By capitalizing on its high temporal resolution, the present study was conducted using ERP recording to investigate the time course of syllabic and tonal encoding in Cantonese speech production and to verify the different views about tonal encoding (e.g., early vs. late encoding) proposed in the literature.

The present study was conducted to investigate how tone is processed together with atonal syllable in Cantonese spoken word production using the PWI task with concurrent ERP recording. The target and distractor (both were disyllabic words in Cantonese) shared either the same word-initial tonal syllable (Tonal Syllable related), the same word-initial atonal syllable (Atonal Syllable related), the same tone in their word-initial syllable position (Tone alone related), or were unrelated to each other (Unrelated Control). Based on the results of previous behavioural studies, significant priming effects on picture naming latency were expected in the Tonal Syllable related and Atonal Syllable related conditions but null effects were expected in the Tone alone related condition [21–23]. In addition, the size of Tonal Syllable priming was expected to be larger than that of the Atonal Syllable priming. The contrast between Tonal Syllable related and Atonal Syllable related conditions would provide us important insights of how tone and atonal syllable are processed with each other. Based on recent ERP findings of Cantonese and Mandarin phonological planning [34,35,37], it was expected that significant ERP effects would be found in Tonal Syllable and Atonal Syllable related conditions. More importantly, it was of particular interest to see whether and how the ERP effects associated with Tonal Syllable priming and Atonal Syllable priming would diverge from each other, similar to the distinctive behavioural results previously reported. In addition, if the null effect of tone alone reported in the previous studies was due to the reason of lack of task sensitivity, a significant effect of tone alone might be observed in the ERP data despite of a null behavioural result. In further addition, recent evidence exists suggesting that by analyzing not only the ERP data time-locked to the stimulus onset (i.e., the stimulus-locked approach) but also the ERP data time-locked to the naming response onset (i.e., the response-locked approach) would produce a clearer picture of the entire speech production processes [38]. The ERP data obtained in the present study were therefore submitted for stimulus- and response-locked analyses, with the former assumed to be more sensitive to the early stages (e.g., stimulus perception and response decision/selection) of speech planning and the latter more sensitive to the late stages (e.g., response decision/selection and execution) of speech preparation [38,39].

The hypotheses of the present study are presented in the following. There is considerable behavioural and electrophysiological evidence in the literature to suggest that the atonal syllable is the first retrievable (proximate) phonological planning unit in Chinese speech production, therefore in the present Cantonese Chinese study, the onset of the ERP effect associated with atonal syllable priming will be used as a reference to indicate the time of when the early stage of phonological encoding occurs. According to the late encoding view [17,19,20],

phonological encoding starts with the retrieval of the atonal syllable, and the tone (as specified by a frame-like structure) will come into play and associate with the segmental contents only at a later stage. Accordingly, the initial stage of phonological encoding is dominated by the retrieval of the atonal syllable. Consequently, similar ERP effects would be expected in the Tonal Syllable related and Atonal Syllable related conditions at an early ERP time window, as the atonal syllable unit is similarly primed in these two conditions. In addition, significant contrasts between these two conditions would be expected at a later time window to reflect the joint effect of concurrent tonal and syllable priming which was present only in the Tonal Syllable related condition but not in the Atonal Syllable related condition. In contrast, according to the early encoding view [13], tonal and segmental contents are concurrently selected and encoded in an interactive manner at the beginning of phonological encoding. Consequently, distinctive ERP effects would be expected between Tonal Syllable related and Atonal Syllable related conditions at an early time window (right at the beginning of phonological encoding), with a more robust ERP effect obtained in the Tonal Syllable related condition than that in the Atonal Syllable related condition. In further addition, the importance of including the Tone alone related condition is two-fold. One is to examine whether the null behavioural effect of the tone alone condition previously reported was due to task insensitivity. If that is the case, a significant ERP effect, despite of the null behavioural result, might be observed in the Tone alone related condition of this study. The other is to further investigate the interactivity between tonal and syllabic processing. If a more robust ERP effect is observed in the Tonal Syllable related condition than that in the Atonal Syllable related condition, and the difference is due to the additive effects of tonal and syllable priming (i.e., tone + atonal syllable effects), then a significant effect would also be expected in the Tone alone related condition. If, however, the difference between Tonal Syllable and Atonal Syllable conditions is due to the interactivity between tonal and syllabic encoding, a stronger ERP effect might be observed in the Tonal Syllable related than the Atonal Syllable related condition, despite a null ERP effect of the Tone alone related condition (similar to the predicted behavioural results).

## 2. Method

### 2.1 Participants

Thirty-four native Cantonese-speaking undergraduates from the City University of Hong Kong (8 males; mean age = 21.0 ± 1.3 years) participated. Participants were all neurologically healthy, without a prior history of speech or hearing impairment, and had normal or corrected-to-normal vision. The experiment was approved by the Human Subjects Ethics Sub-Committee of the Research Committee of the City University of Hong Kong. Informed consent was obtained from each participant before the experiment. Each participant was tested individually. Thirty-three participants were right-handed and one was left-handed according to their scores on the Edinburgh inventory for handedness [40]. Participants were remunerated with HKD$100 (~USD 13) per hour for their participation.

### 2.2 Materials and apparatus

Forty-eight black-on-the-white line drawings of common objects adopted in previous related studies [34,41] were used as stimuli. Each picture had a disyllable Cantonese name and paired with three phonologically related auditory distractors (which were all Cantonese disyllabic words) to form three phonologically related distractor conditions. Each picture name and its corresponding distractor shared either the same word-initial syllable together with the same tone (i.e., Tonal-syllable related), the same word-initial syllable but without the same tone (i.e., Atonal-syllable related), or the same tone in the word-initial syllable position only (i.e., Tone

alone related). The three sets of distractors were closely matched in their number of homophones, as well as the frequency (type or token frequency) of their first and second syllables [42,43], $Fs < 1.70$. The auditory distractors used in this study were recorded by a female native Cantonese speaker digitalized with a sampling rate of 20k Hz. The duration of the auditory distractors was measured using the software Audacity, and the mean durations of the Tonal Syllable related, Atonal Syllable related, and Tone alone related distractors were 749±101 ms, 740 ±85 ms, and 742±106 ms, respectively. The mean duration of the three types of distractors was not significantly different from each other, $F < 0.2$, $p > 0.8$. In addition, an independent group of 16 native Cantonese speakers (who were not participating in the main ERP experiment) were recruited to be raters to rate the subjective word frequency of all the distractor words on a 7-point Likert Scale ranging from 1 (Very infrequently use or perceive) to 7 (Very frequently use or perceive). The mean rating scores of the Tonal Syllable related, Atonal Syllable related, and Tone alone related distractors were 6.23±0.5, 6.29±0.47, and 6.27±0.41, respectively, and the difference was not significant, $F < 0.2$, $p > 0.8$. Furthermore, three corresponding unrelated control conditions were created by re-pairing the targets and distractors within the same condition so that the target and distractor in a pair were unrelated to each other. Consequently, six distractor conditions were included, namely tonal syllable related, tonal syllable unrelated control, atonal syllable related, atonal syllable unrelated control, tone related, and tone unrelated control conditions. In addition, the target and distractor in each pair were orthographically unrelated in their written form and were not semantically related in any obvious way. Samples of stimuli are shown in Fig 1. A list of the target and distractor stimuli is provided in S1 Appendix. It has to be noted that some of the di-syllabic targets in the present study were composed by two syllables sharing the same tone. This was due to the constraints in material construction. The consensus in the past literature was that the primary planning unit in Cantonese and Mandarin Chinese speech production is the atonal syllable (e.g. [17,35]). Therefore, cautions were taken to ensure that each target was composed by two different atonal syllables. Future research is required to examine whether tonal priming occurs between the two syllables within a di-syllabic target. Nevertheless, even if this within-target priming effect exists, such effect should be constant across conditions, as the same set of targets was used across the different conditions in this study.

Electroencephalogram (EEG) was recorded from 64 scalp sites by the eego™ mylab system (ANT Neuro). Horizontal and vertical electrooculogram (EOG) was recorded for ocular artifact reduction. Both EEG and EOG were digitized at 1000Hz. The impedance of each electrode was kept below 10kΩ. EEG data were re-referenced offline from CPz to the average of all channels. Band-pass filter of 0.05-30Hz was applied after re-referencing. Then, the EEG data were segmented into epochs of 200 ms pre- to 500 ms post-stimulus onset (for stimulus-locked analyses), which were baseline corrected by the average EEG signals within the 200 ms preceding the stimulus onset. Afterwards, ocular artifact reduction was performed. 14.4% of the total epoch were discarded from further EEG analyses due to mistrigger (2.9%) or incorrect response (1.6%), response time outlier (<500ms or >1500ms; 0.7%) or excessive artifact (>±120μV; 11.2%). Data from six participants were discarded from ERP data analyses due to the abnormal noise found in the data and high discard rate (>35% of trials).

In addition, similar analyses were performed with the data from all participants but time-locked to the vocal response onset. In this response-locked analysis, the EEG signals obtained within the 500-ms time period preceding response onset with a baseline corresponding to the 200-ms time window between 2000 and 1800 ms prior to the response onset were used for analysis. The baseline time window was chosen in such a way to ensure that all the signals for baseline measurement were coming from the time when the fixation was shown on the screen

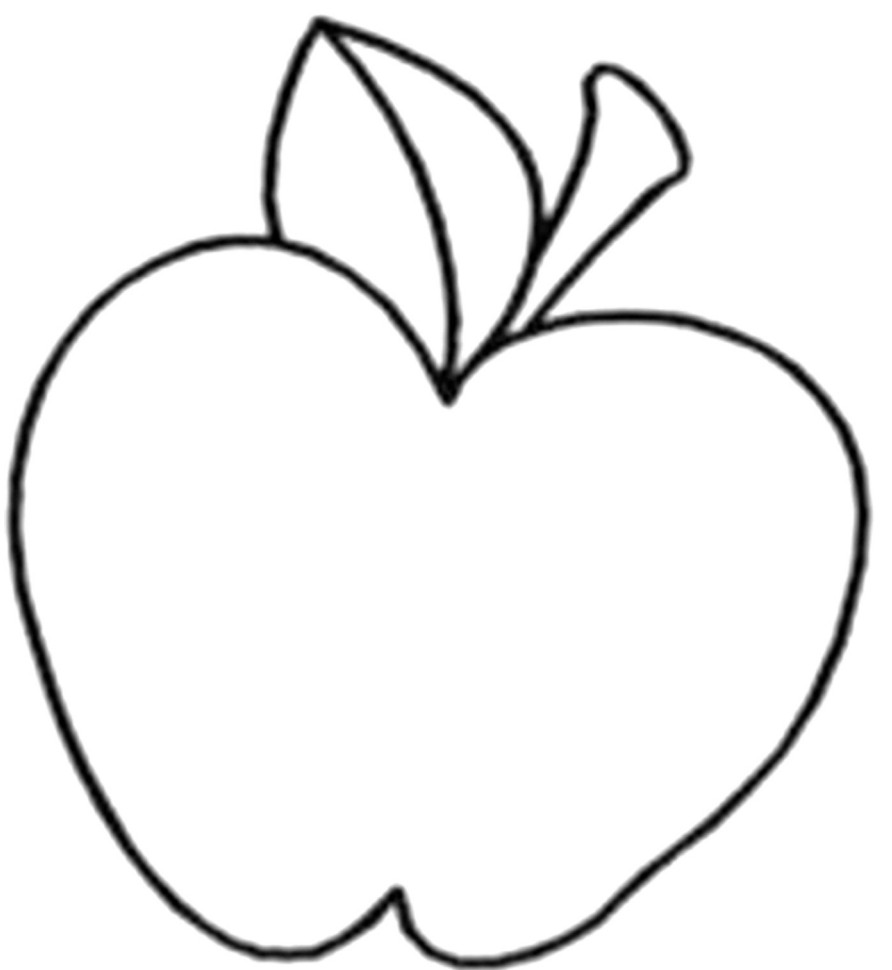

**Fig 1. Sample of experimental stimulus.**

| Picture: | Apple [/**ping4** gwo2/] |
| --- | --- |
| Distractor | |
| Tonal-syllable related: | /**ping4** zing6/ "calm" |
| Tonal-syllable unrelated control: | /si1 wai4/ "cognition" |
| Atonal-syllable related: | /**ping3** jam1/ "pinyin" |
| Atonal-syllable unrelated control: | /sau6 ming6/ "life span" |
| Tone related: | /**ceon4** si6/ "patrol" |
| Tone unrelated control: | /zyut6 baan2/ "limited edition" |

(which was consistent across all distractor conditions). For the response-locked analysis, epoch rejection rate due to excessive artifact (>±120μV) was 15.7% and data from eight participants were discarded from ERP data analyses due to high discard rate (>35% of trials).

ERP data were computed separately for each distractor condition. ERP waveforms from proximal electrodes were further averaged into six regions of interest (ROIs) following the practice of previous ERP studies on overt speech production [34,35]: left-anterior (F5, F7, FC5), middle-anterior (Fz, FCz, Cz), right-anterior (F6, F8, FC6), left-posterior (P5, P7, CP5), middle-posterior (Pz, POz, Oz) and right-posterior (P6, P8, CP6) regions.

## 2.3 Design and procedure

The experiment adopted a 3 (Distractor Type: tonal syllable vs. atonal syllable vs. tone only) x 2 (Target-Distractor Relatedness: related vs. unrelated) within-participants design. Each participant named each of the 48 pictures under each of the six distractor conditions once, comprising in total 288 (48 pictures x 6 distractor conditions) experimental trials. Six blocks of 48 trials each were formed such that no picture appeared more than once within a block. At the beginning of each block, two filler trials (as warm up trials) comprised of materials that were not included in the subsequent experimental trials and analyses were presented first. The six distractor conditions were intermixed and equiprobable in each block, and the trial order was randomized. The block order was randomized across participants. The experiment consisted of a learning phase and a testing phase. In the learning phase, participants were shown on a computer screen each of the pictures along with their names. The picture names were then tested immediately afterward to check if participants were able to name the pictures correctly. In the subsequent actual testing phase, each trial started with a fixation presented at the center of the screen for 1200 ms, followed by a 300-ms blank. The target picture and distractor were then presented simultaneously and the picture stayed on the screen for 3000 ms or until a naming response was detected. Participants were asked to name aloud the target picture as accurately and quickly as possible, and to ignore an accompanying auditory distractor. The onset of naming response was detected by a microphone (which was placed approximately 10 centimeters in front of the participant's mouth) connecting to the same computer for stimulus presentation. The inter-trial-interval varied from 2200 ms to 2500 ms randomly. Prior to the start of the experiment, participants learned and named 10 pictures which were not included in the subsequent experimental session for practice. E-Prime 2.0 software was used to present the pictures and auditory distractors. Pictures were shown with a size of 6 cm × 6 cm (approximately 5˚ × 5˚ in visual angle) on the computer screen with a black background. In addition to the EEG recording equipment, participants wore a pair of in-ear headphone for distractor presentation throughout the experiment. The entire experiment lasted for approximately 1.5 hours.

## 3. Results

### 3.1 Behavioural

Incorrect (1.6%) or mis-triggered (2.9%) responses and responses shorter or longer than 3 standard deviations (1.7%) from either the participants' or items' mean were excluded from reaction time analyses. Table 1 shows the participants' mean naming latencies across conditions.

The remaining naming latency data were then inverse transformed (invRT = -1000/RT) and submitted to linear mixed-effect modeling (LMEM) [44] implemented in R Version 4.10 [45]. The lmerTest package [46] was used to conduct $F$ tests on the fixed effects and calculate $p$ values with Satterthwaite approximation to prevent Type 1 error [47]. Distractor Type (tonal syllable vs. atonal syllable vs. tone alone only), Target-Distractor Relatedness (related vs. unrelated), and their interaction were fixed effects, whereas participants and items (picture stimuli) were random effects. Block order and trial order within a block were entered to control for possible confounds. The maximum model that converged was examined [48] and the final model included by-participant and by-item random intercepts, and their random slopes for Target-Distractor Relatedness: invRT ~ DistractorType * Relatedness + Block + Trial + (1 + Relatedness |Item) + (1+ Relatedness |Participant). Results of $F$ tests on the fixed effects using Satterthwaite approximation are shown in Table 2.

**Table 1. Participants' mean naming latencies (in ms) across conditions.**

| Distractor Type | | Related | Unrelated (Control) | Difference |
|---|---|---|---|---|
| Tonal Syllable | Mean | 692 | 772 | -80 |
| | SD | 138 | 158 | |
| Atonal Syllable | Mean | 704 | 759 | -55 |
| | SD | 142 | 144 | |
| Tone alone | Mean | 758 | 757 | 1 |
| | SD | 146 | 147 | |

*Note.* SD: Standard deviation. Difference: Related–Unrelated.

Since a significant Distractor Type x Relatedness interaction was observed, the simple main effect of Relatedness was then analyzed separately for each Distractor Type condition: invRT ~ Relatedness + Block + Trial + (1 + Relatedness |Item) + (1+ Relatedness |Participant). The unrelated control condition was treated as the baseline. The regression coefficients (*b*), standard errors (*SE*), and *p* values are reported in Table 3. The main effect of Relatedness was significant in the Tonal Syllable and the Atonal Syllable conditions but was not in the Tone alone overlap condition.

An additional analysis was conducted to directly compare the size of the priming effects observed in the Tonal Syllable related and the Atonal Syllable related conditions. To this end, only the data from these two distractor type conditions were included in the analysis. Similar to the first round analysis in the above, Distractor Type, Relatedness, and their interaction were treated as fixed effects, together with Block and Trial. And participants and items were random effects. Results of *F* tests on the fixed effects for this additional analysis are shown in Table 4.

Results of the additional analysis showed a significant effect of Relatedness but a null effect of Distractor Type. More importantly, a significant Distractor Type x Relatedness interaction was observed, indicating that the priming effect observed in the Tonal Syllable condition was significantly larger than that in the Atonal Syllable condition.

## 3.2 ERP

**3.2.1 Stimulus-locked analysis.** The grand average ERPs across ROIs and conditions by stimulus-locked analyses (time-locked to the stimulus onset until 500-ms post-stimulus) are displayed in Figs 2–4. Mean ERP amplitude values were calculated over five successive time windows of 100 ms immediately following stimulus onset in each condition. For each 100-ms time window, a repeated-measures 4-way ANOVA was conducted on the mean ERP amplites with Distractor Type (TYPE: tonal syllable related, atonal syllable related or tone alone related), Target-Distractor Relatedness (RELATEDNESS: related vs. unrelated), Anteriority

**Table 2. Results of *F* tests on the naming latency data from all three Distractor Type conditions.**

| | Sum of Square | Mean Square | df numerator | df denominator | F | p | |
|---|---|---|---|---|---|---|---|
| Block | 20.45 | 20.45 | 1 | 8709.1 | 433.04 | < .001 | *** |
| Trial | 8.66 | 8.66 | 1 | 9083 | 183.33 | < .001 | *** |
| Relatedness | 3.88 | 3.88 | 1 | 51.1 | 77.04 | < .001 | *** |
| DistractorType | 6.58 | 3.29 | 2 | 9028.3 | 65.36 | < .001 | *** |
| Relatedness x DistractorType | 10.95 | 5.47 | 2 | 9028.7 | 108.75 | < .001 | *** |

**Table 3. Simple main effects of Relatedness in each Distractor Type condition.**

| Distractor Type | Fixed effect | b | SE | t | p | |
|---|---|---|---|---|---|---|
| Tonal Syllable | (Intercept) | -1.31 | 0.034 | -38.33 | < 0.001 | *** |
| | Block | -0.032 | 0.002 | -13.17 | < 0.001 | *** |
| | Trial | 0.003 | 0.0003 | 8.85 | < 0.001 | *** |
| | Relatedness | -0.166 | 0.018 | -9.46 | < 0.001 | *** |
| Atonal Syllable | (Intercept) | -1.32 | 0.034 | -38.93 | < 0.001 | *** |
| | Block | -0.03 | 0.002 | -13.2 | < 0.001 | *** |
| | Trial | 0.002 | 0.0003 | 6.95 | < 0.001 | *** |
| | Relatedness | -0.12 | 0.015 | -7.64 | < 0.001 | *** |
| Tone | (Intercept) | -1.35 | 0.034 | -40.41 | < 0.001 | *** |
| | Block | -0.022 | 0.002 | -9.97 | < 0.001 | *** |
| | Trial | 0.002 | 0.0003 | 7.56 | < 0.001 | *** |
| | Relatedness | 0.0009 | 0.009 | 0.102 | 0.919 | |

(ANT: anterior vs. posterior regions), and Laterality (LAT: left, middle, or right ROIs) as within-participants factors. As multiple ANOVA tests were performed, Bonferrioni procedure was adopted to control for Type I error with the alpha value adjusted to .01 level (.05 was divided by 5 times of ANOVAs). Table 5 lists the *F* and *p* values of the effects involving the factor Target-Distractor Relatedness.

Significant Relatedness x Laterality interactions were found in the time windows of 300–400 ms and 400–500 ms post-stimulus onset (*ps* < .01). The main effect of Relatedness and all other interactions involving the factor Relatedness were not significant (at alpha = .01 level) in any of the five time windows. The Relatedness x Laterality interactions suggest that the effect of relatedness was not the same across the three levels of laterality (left, middle, and right ROIs) in the last two 100-ms time windows. The non-significant (*ps* > .01) Relatedness x Distractor Type x Laterality three-way interaction suggests that a similar pattern of Relatedness x Laterality interaction existed in all three distractor type conditions. However, a closer inspection of the ERP waveforms (Figs 2–4) does not seem to suggest that the effect of relatedness is similarly noticeable across the three distractor types. To take a closer look into the Relatedness x Laterality interactions and to see if a similar pattern existed in all three Distractor Type conditions (which is of particular interest of the present study), further *t*-tests were performed to examine the simple main effects of relatedness at different levels of laterality across different distractor conditions during the time windows of 300–400 ms and 400–500 ms post-stimulus. For each of the two significant Relatedness x Laterality interactions (one in the 300–400 ms time window and another in the 400–500 ms time window), nine follow up *t*-tests were performed to examine the source of interaction. To control for Type I error with Bonferroni correction, the alpha value was adjusted to .0055 (0.05/9 = 0.0055) in the follow up *t*-tests. Table 6 shows the results of the follow up comparisons.

**Table 4. Results of F tests on the naming latency data from the Tonal Syllable and the Atonal Syllable conditions only.**

| | Sum of Square | Mean Square | df numerator | df denominator | F | p | |
|---|---|---|---|---|---|---|---|
| Block | 16.38 | 16.38 | 1 | 5940.6 | 344.12 | < .001 | *** |
| Trial | 6 | 6 | 1 | 6021.7 | 126.15 | < .001 | *** |
| Relatedness | 4.5 | 4.5 | 1 | 48.2 | 94.55 | < .001 | *** |
| DistractorType | 0.069 | 0.069 | 1 | 5974.6 | 1.44 | 0.23 | |
| Relatedness x DistractorType | 0.867 | 0.867 | 1 | 5975.1 | 18.21 | < .001 | *** |

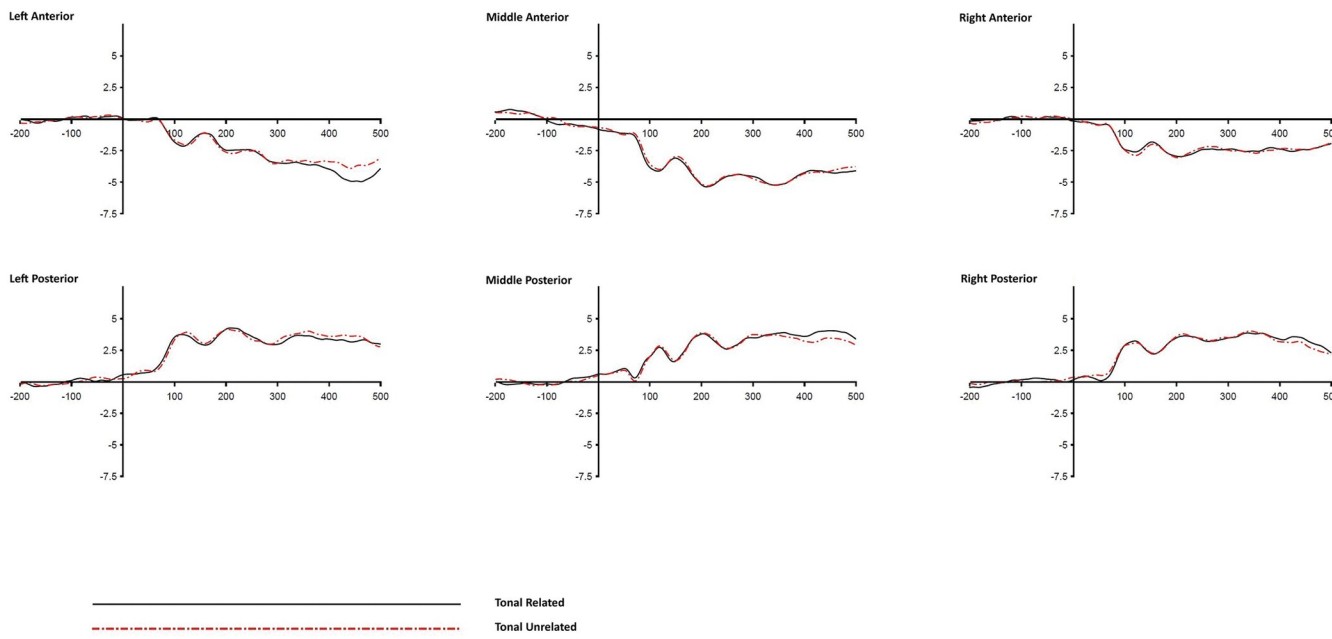

**Fig 2. Grand averaged ERP waves of the Tonal Syllable condition across different ROIs (stimulus-locked).**

Significant effects of Relatedness ($ps < .0055$) were found in the Tonal Syllable related and the Atonal Syllable related conditions in the time window of 400–500 ms but the effects were confined to the left ROIs only, which accounts for the Relatedness x Laterality interaction observed in the same time window in the ANOVA test. Although the Relatedness x Laterality interaction was also significant in the 300–400 ms time window, follow up $t$-tests do not show

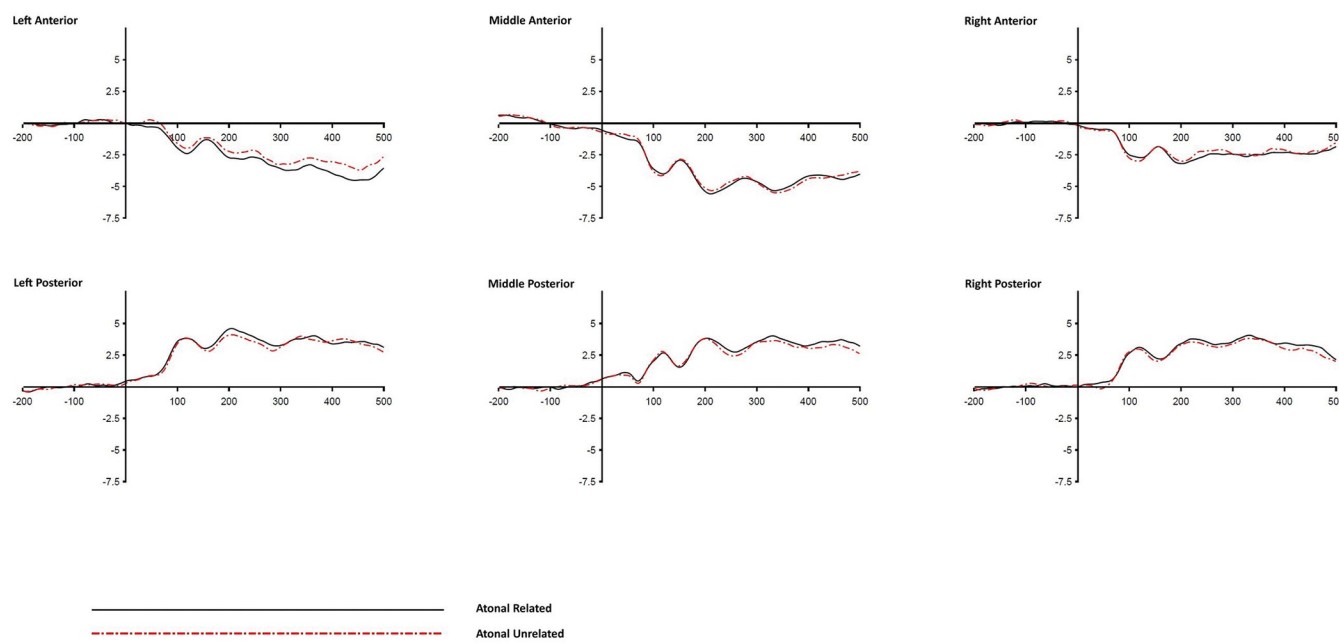

**Fig 3. Grand averaged ERP waves of the Atonal Syllable condition across different ROIs (stimulus-locked).**

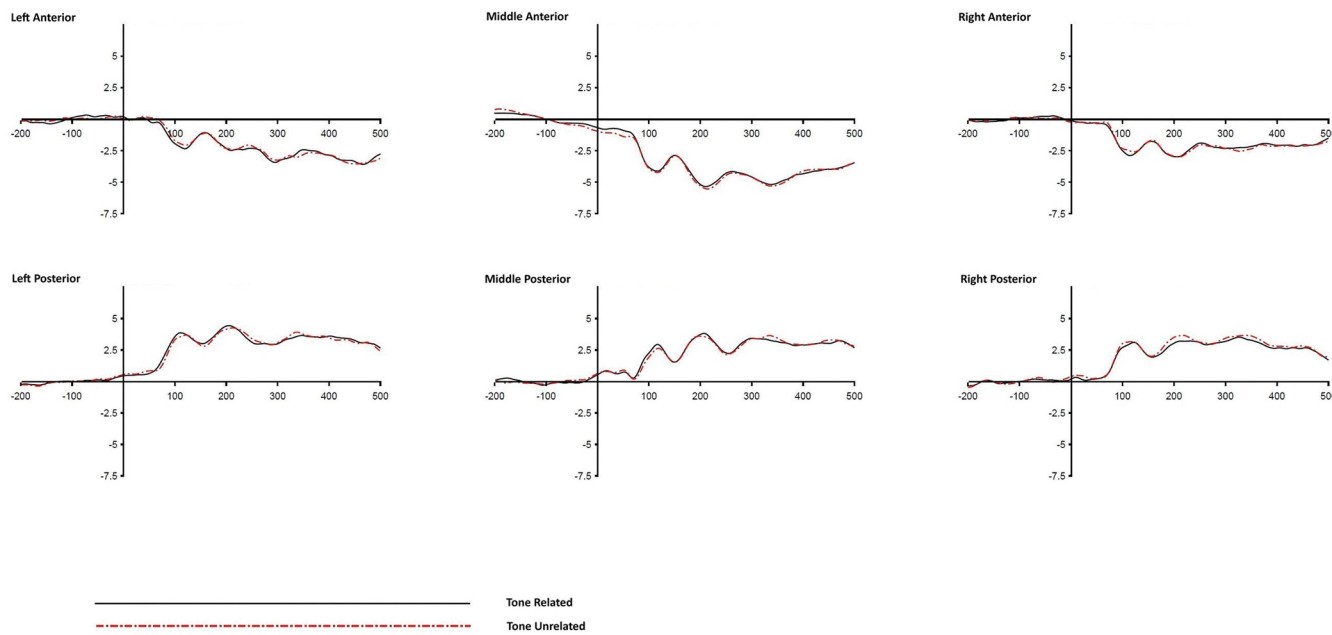

**Fig 4. Grand averaged ERP waves of the Tone alone condition across different ROIs (stimulus-locked).**

a clear and robust effect of relatedness in any of the three Distractor Type conditions during this time window. More importantly, the Tone alone related condition did not show any significant effects (all $ts < 1$, and $ps > .3$) in the follow up $t$-tests, indicating that the effects of Relatedness were not all the same across the three Distractor Type conditions.

To further examine the degree of which the present evidence supports a null effect of the Tone alone related condition (i.e., the null hypothesis, $H_0$), instead of merely an absence of evidence in favor of the alternative hypothesis (i.e., $H_1$), Bayes factors ($BF_{10}$) were calculated using the software version 0.18 [49]. Corresponding paired samples t-tests were conducted to examine the effect of Relatedness in each level of Laterality and in each Distractor Type in the time windows of 300–400 ms and 400–500 ms post-stimulus using the Bayesian approach. The resultant $BF_{10}$ values are shown in Table 6. Consistent with the above classical analysis with

**Table 5. Summary of ANOVAs on the mean ERP amplitudes in each 100-ms post-stimulus time window (only the main effect of and interactions involving the factor Target-Distractor Relatedness are listed).**

| | | | Post-stimulus time windows | | | | | | | | | |
|---|---|---|---|---|---|---|---|---|---|---|---|---|
| | | | 0–100 ms | | 100–200 ms | | 200–300 ms | | 300–400 ms | | 400–500 ms | |
| Effects of Relatedness | *df1* | *df2* | F | p | F | p | F | p | F | p | F | p |
| Relatedness | 1 | 27 | 0.16 | .696 | 0.63 | .436 | 0.92 | .347 | 1.69 | .204 | 2.83 | .104 |
| Relatedness x Type | 2 | 26 | 0.14 | .870 | 0.20 | .821 | 0.01 | .991 | 0.17 | .847 | 1.31 | .288 |
| Relatedness x Ant | 1 | 27 | 0.60 | .444 | 0.23 | .634 | 1.37 | .253 | 0.003 | .958 | 5.80 | .023 |
| Relatedness x Lat | 2 | 26 | 1.24 | .307 | 0.07 | .929 | 0.84 | .445 | 5.90 | .008* | 5.77 | .008* |
| Relatedness x Type x Ant | 2 | 26 | 0.43 | .658 | 0.34 | .715 | 1.64 | .213 | 0.94 | .404 | 1.56 | .230 |
| Relatedness x Type x Lat | 4 | 24 | 2.15 | .106 | 3.36 | .026 | 0.58 | .682 | 1.04 | .409 | 2.38 | .080 |
| Relatedness x Ant x Lat | 2 | 26 | 3.35 | .051 | 2.15 | .137 | 1.95 | .162 | 1.36 | .273 | 2.47 | .104 |
| Relatedness x Type x Ant x Lat | 4 | 24 | 3.06 | .036 | 1.26 | .315 | 0.47 | .760 | 0.65 | .630 | 0.27 | .894 |

*Note.* * Significant at alpha = .01 level. Type: Type of phonological overlap (Tonal syllable vs. Atonal syllable vs. Tone alone); Ant: Anteriority, Lat: Laterality.

**Table 6. Effects of relatedness at different levels of laterality across different distractor conditions during the time windows of 300–400 ms and 400–500 ms post-stimulus onset.**

| | | | Laterality | | | | | | | | |
| | | | Left | | | Middle | | | Right | | |
| Time window | Distractor Type | df | t | p | $BF_{10}$ | t | p | $BF_{10}$ | t | p | $BF_{10}$ |
|---|---|---|---|---|---|---|---|---|---|---|---|
| 300–400 ms | Tonal Syllable | 27 | -1.892 | 0.069 | 0.95 | 1.177 | 0.249 | 0.38 | 0.507 | 0.617 | 0.23 |
| | Atonal Syllable | 27 | -2.514 | 0.018 | 2.79 | 2.698 | 0.012 | 3.99 | 0.104 | 0.918 | 0.20 |
| | Tone alone | 27 | -0.097 | 0.923 | 0.20 | -0.34 | 0.737 | 0.21 | -0.407 | 0.687 | 0.22 |
| 400–500 ms | Tonal Syllable | 27 | -3.214 | 0.003* | 11.65 | 1.999 | 0.056 | 1.13 | 1.146 | 0.262 | 0.36 |
| | Atonal Syllable | 27 | -3.143 | 0.004* | 9.99 | 1.766 | 0.089 | 0.79 | 1.166 | 0.254 | 0.37 |
| | Tone alone | 27 | 0.996 | 0.328 | 0.32 | -0.466 | 0.645 | 0.22 | -0.143 | 0.887 | 0.20 |

*Note.* * Significant at alpha = .0055 level. $BF_{10}$ values below 0.33: Moderate evidence for $H_0$; Between 0.33 and 1: Weak evidence for $H_0$; between 1 and 3: Weak evidence for $H_1$; between 3 and 10: Moderate evidence for $H_1$; between 10 and 30: Strong evidence for $H_1$.

the p values, moderate to strong evidence in favor of $H_1$ (i.e., an effect does exist) was found in the Tonal Syllable related and Atonal Syllable related conditions in the 400–500 ms time window post-stimulus (for the interpretation of the Bayes factors $BF_{10}$, please refer to [50,51]). Notably, the $BF_{10}$ values of the Tone alone related condition were all smaller than 0.33, indicating moderate evidence in favor of $H_0$ (i.e., an effect does not exist) in stimulus-locked analysis.

**3.2.2 Response-locked analysis.** The grand average ERPs time-locked to the response onset (i.e., the 500-ms time period before response onset, which was based on the response latency) as a function of ROIs and distractor conditions are shown in Figs 5–7. Mean ERP amplitudes were calculated for each condition in each of the five successive time windows of 100 ms preceding reponse onset. Similar to the stimulus-locked analysis, five ANOVAs (one in each 100-ms pre-response time window) with Distractor Type, Relatedness, Anteriority, and Laterality as within-participants factors were conducted on the mean ERP amplitude values.

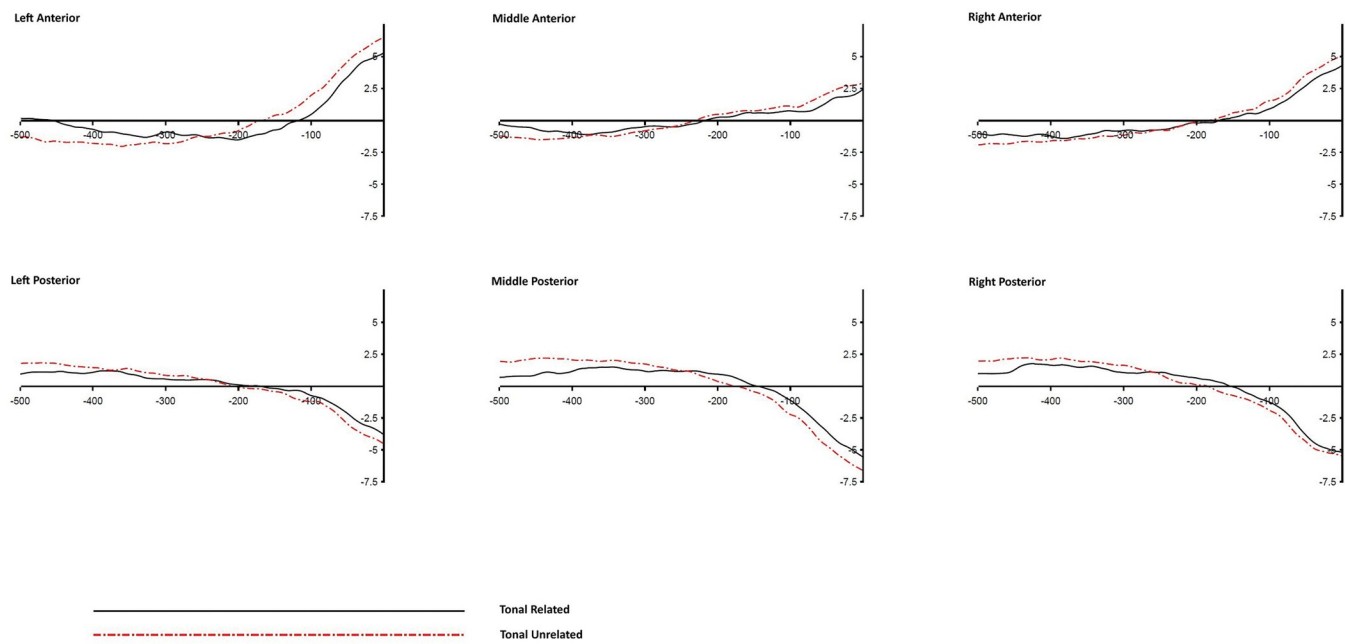

**Fig 5. Grand averaged ERP waves of the Tonal Syllable condition across different ROIs (response-locked).**

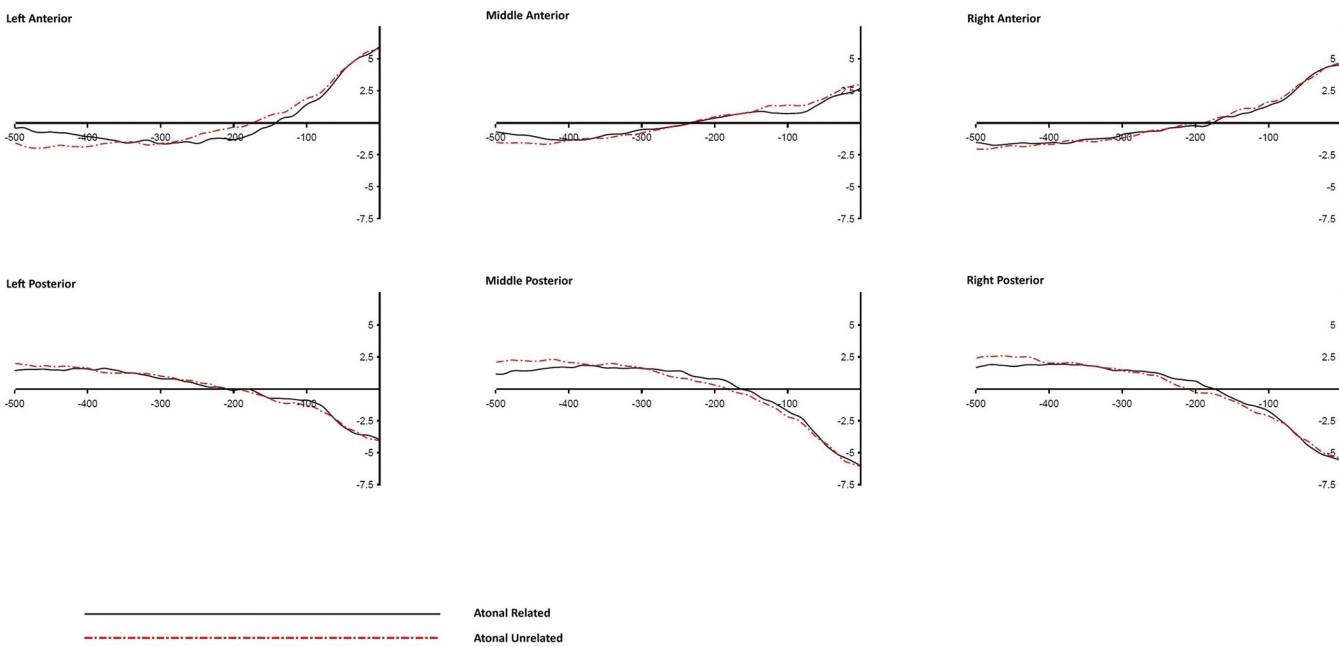

**Fig 6. Grand averaged ERP waves of the Atonal Syllable condition across different ROIs (response-locked).**

Similarly, the Bonferrioni procedure was adopted and the alpha value was adjusted to .01 level (0.05/5) to control for Type I error. Table 7 lists the *F* and *p* values of the effects involving the factor Target-Distractor Relatedness in response-locked analysis.

A significant Relatedness x Anteriority interaction was found in the time windows of -500 to -400 ms, -200 to -100 ms, and -100 to 0 ms ($ps < .01$). Similarly, a significant Relatedness x

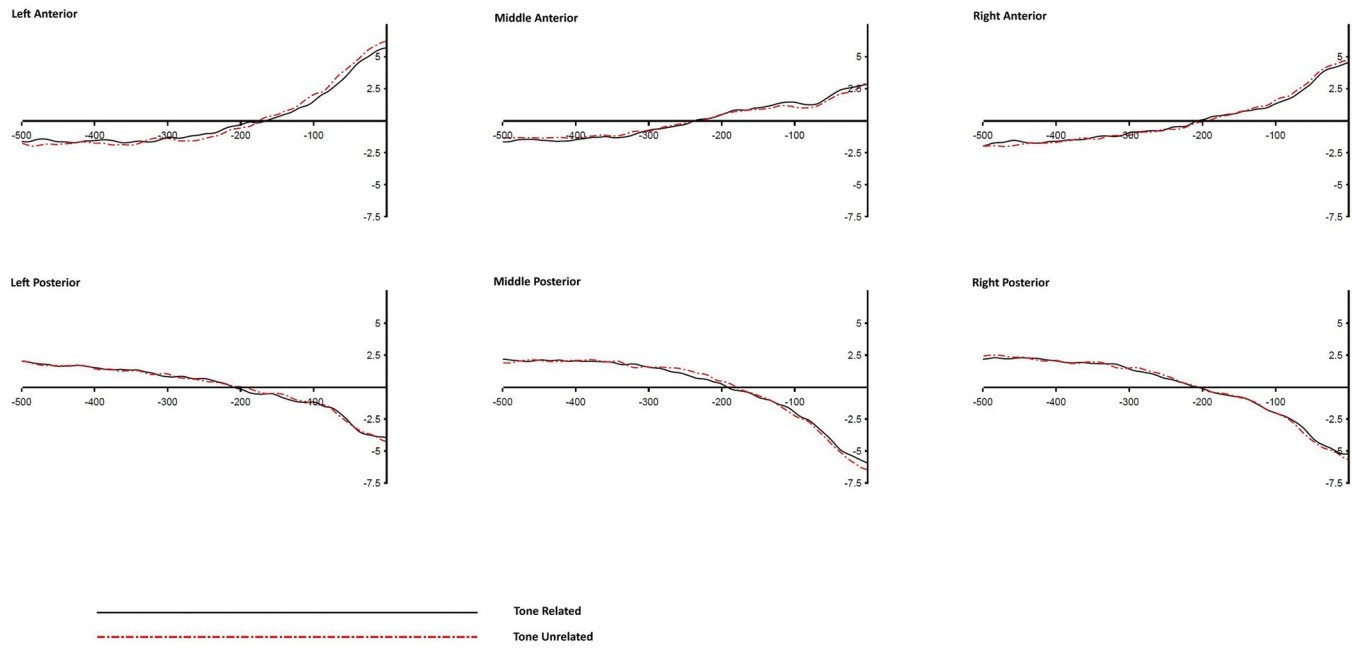

**Fig 7. Grand averaged ERP waves of the Tone alone condition across different ROIs (response-locked).**

**Table 7. Summary of ANOVAs on the mean ERP amplitudes in each 100-ms pre-response time window (only the main effect of and interactions involving the factor Target-Distractor Relatedness are listed).**

| | | | Pre-response time windows | | | | | | | | | |
| --- | --- | --- | --- | --- | --- | --- | --- | --- | --- | --- | --- | --- |
| | | | -500 to -400 ms | | -400 to -300 ms | | -300 to -200 ms | | -200 to -100 ms | | -100 to 0 ms | |
| Effects of Relatedness | df1 | df2 | F | p | F | p | F | p | F | p | F | p |
| Relatedness | 1 | 25 | 2.57 | .121 | 1.04 | .317 | 0.61 | .441 | 3.12 | .089 | 3.28 | .082 |
| Relatedness x Type | 2 | 24 | 0.09 | .912 | 0.57 | .574 | 0.47 | .631 | 0.47 | .634 | 1.99 | .159 |
| Relatedness x Ant | 1 | 25 | 27.57 | < .001* | 5.44 | .028 | 0.01 | .945 | 17.58 | < .001* | 17.67 | < .001* |
| Relatedness x Lat | 2 | 24 | 17.40 | < .001* | 2.82 | .080 | 3.00 | .069 | 7.16 | .004* | 3.66 | .041 |
| Relatedness x Type x Ant | 2 | 24 | 12.27 | < .001* | 5.60 | .010# | 2.18 | .135 | 8.80 | .001* | 10.50 | .001* |
| Relatedness x Type x Lat | 4 | 22 | 1.47 | .245 | 0.72 | .588 | 4.34 | .010* | 1.94 | .140 | 2.32 | .089 |
| Relatedness x Ant x Lat | 2 | 24 | 6.17 | .007* | 1.08 | .357 | 0.22 | .806 | 4.90 | .016 | 4.43 | .023 |
| Relatedness x Type x Ant x Lat | 4 | 22 | 3.27 | .030 | 1.48 | .243 | 0.66 | .626 | 2.57 | .066 | 3.67 | .020 |

*Note.* * Significant at alpha = .01 level. # Marginal significant ($p$ = .01012).

Laterality interaction was found in the time windows of -500 to -400 ms, and -200 to -100 ms, ($ps < .01$). However, these two-way interactions were superseded by several higher order three-way interactions including the Relatedness x Distractor Type x Anteriority which was found significant (or marginal significant) in the time windows of -500 to -400 ms, -400 to -300 ms, -200 to -100 ms, and -100 to 0 ms. In addition, a significant Relatedness x Distractor Type x Laterality interaction was found in the -300 to -200 ms window, and a significant Relatedness x Anteriority x Laterality interaction was found in the -500 to -400 ms window. The following follow up analyses and discussions will focus on the observed three-way interactions.

To examine the source of the Relatedness x Type x Anteriority interactions, follow up $t$-tests were performed to examine the simple main effects of relatedness at different levels of Anteriority (i.e., anterior vs. posterior ROIs) across different distractor conditions during the time windows of -500 to -400 ms, -400 to -300 ms, -200 to -100 ms, and -100 to 0 ms pre-response onset. For each 100-ms time window, six follow up $t$-tests were performed, and to control for Type I error with Bonferroni procedure, the alpha value was adjusted to .0083 (0.05/6 = 0.0083). Table 8 shows the results of the follow up $t$-test comparisons. Significant effects of Relatedness were found in the Tonal Syllable related condition in all four time windows and in both anterior and posterior ROIs. The effects of Relatedness were also significant in the Atonal Syllable related condition in the time windows of -500 to -400 ms and -200 to -100 ms (and in both anterior and posterior ROIs), but were not in the time windows of -400 to -300 ms and -100 to 0 ms pre-response onset. Null effects of Relatedness were found in the Tone alone related condition.

The effect of Relatedness in each level of Anteriority and Distractor Type condition, and in each of the four above mentioned time windows, was also examined using the Bayesian approach, and the corresponding $BF_{10}$ values in different conditions are shown in Table 8. Evidence in favor of $H_1$ (with the $BF_{10}$ values ranging from 31.13 [very strong evidence] to 9914.94 [extreme evidence]) was found in the Tonal Syllable related condition in all four time windows. Evidence in favor of $H_1$ (with the $BF_{10}$ values ranging from 6.93 [moderate evidence] to 165.2 [extreme evidence]) was also found in the Atonal Syllable related condition but the evidence was limited to the time windows of -500 to -400 ms and -200 to -100 ms only. Moderate ($BF_{10}$ below 0.33) to weak ($BF_{10}$ between 0.33 and 1) evidence in favor of $H_0$ was found in the Atonal Syllable related condition in the time windows of -400 to -300 ms and -100 to 0 ms. Importantly, for the Tone alone related condition, the corresponding $BF_{10}$ values were almost

**Table 8. Effects of relatedness at different levels of anteriority across different distractor conditions during the time windows of -500 to -400 ms, -400 to -300 ms, -200 to -100 ms, and -100 to 0 ms pre-response onset.**

| | | | Anteriority | | | | | |
| --- | --- | --- | --- | --- | --- | --- | --- | --- |
| | | | Anterior | | | Posterior | | |
| Time window | Distractor Type | df | t | p | $BF_{10}$ | t | p | $BF_{10}$ |
| -500 to -400 ms | Tonal Syllable | 25 | 6.168 | < .001* | 9914.94 | -5.601 | < .001* | 2658.19 |
| | Atonal Syllable | 25 | 4.413 | < .001* | 165.20 | -4.085 | < .001* | 77.57 |
| | Tone alone | 25 | 0.519 | 0.608 | 0.23 | -0.015 | 0.988 | 0.21 |
| -400 to -300 ms | Tonal Syllable | 25 | 3.744 | < .001* | 35.83 | -3.681 | .001* | 31.13 |
| | Atonal Syllable | 25 | 1.23 | 0.23 | 0.41 | -0.674 | 0.506 | 0.26 |
| | Tone alone | 25 | 0.036 | 0.971 | 0.21 | 0.099 | 0.922 | 0.21 |
| -200 to -100 ms | Tonal Syllable | 25 | -5.441 | < .001* | 1828.66 | 4.874 | < .001* | 483.95 |
| | Atonal Syllable | 25 | -3.483 | .002* | 20.15 | 2.979 | .006* | 6.93 |
| | Tone alone | 25 | -0.152 | 0.88 | 0.21 | -0.705 | 0.487 | 0.26 |
| -100 to 0 ms | Tonal Syllable | 25 | -6.084 | < .001* | 8152.59 | 5.8 | < .001* | 4228.96 |
| | Atonal Syllable | 25 | -1.518 | 0.142 | 0.57 | 0.528 | 0.566 | 0.24 |
| | Tone alone | 25 | -1.333 | 0.195 | 0.46 | 1.464 | 0.156 | 0.53 |

Note. * Significant at alpha = .0083 level. $BF_{10}$ values below 0.33: Moderate evidence for $H_0$; between 0.33 and 1: Weak evidence for $H_0$; between 1 and 3: Weak evidence for $H_1$; between 3 and 10: Moderate evidence for $H_1$; between 10 and 30: Strong evidence for $H_1$; between 30 and 100: Very strong evidence for $H_1$; beyond 100: Extreme evidence for $H_1$.

all below 0.33 (showing moderate evidence in favor of $H_0$) with the only exception in the -100 to 0 ms time window where weak ($BF_{10}$ between 0.33 and 1) evidence in favor of $H_0$ was obtained.

To examine the source of the Relatedness x Type x Laterality interaction in the -300 to -200 ms time window, nine follow up *t*-test comparisons were performed to examine the simple effects of relatedness across different Distractor Type conditions and different levels of Laterality in the same time window. To control for Type I error with Bonferroni procedure, the alpha value was adjusted to .0056 (0.05/9 = 0.0056). A significant effect of Relatedness was found in the Atonal Syllable related condition but the effect was confined to the Left ROIs only (*t*[25] = -4.04, *p* < .001). No other comparisons reached significance at .0056 level (see Table 9).

The effects of Relatedness were also examined using the Bayesian approach (see Table 9). Weak or moderate evidence in favor of $H_0$ was found in the Tonal Syllable related condition in different laterality levels. Whereas very strong evidence in favor of $H_1$ was found in the Atonal Syllable related condition in the left ROIs, though moderate (or mild) evidence in favor of $H_1$ was also found in the middle and right ROIs. For the Tone alone related condition, the

**Table 9. Effects of relatedness at different levels of laterality across different distractor conditions during the time window of -300 to -200 ms pre-response onset.**

| | | | Laterality | | | | | | | | |
| --- | --- | --- | --- | --- | --- | --- | --- | --- | --- | --- | --- |
| | | | Left | | | Middle | | | Right | | |
| Time window | Distractor Type | df | t | p | $BF_{10}$ | t | p | $BF_{10}$ | t | p | $BF_{10}$ |
| -300 to -200 ms | Tonal Syllable | 25 | -0.047 | 0.963 | 0.21 | -0.499 | 0.622 | 0.23 | 1.741 | 0.094 | 0.78 |
| | Atonal Syllable | 25 | -4.041 | < .001* | 70.24 | 2.577 | 0.016 | 3.14 | 2.555 | 0.017 | 3.01 |
| | Tone alone | 25 | 1.759 | 0.091 | 0.80 | -2.317 | 0.029 | 1.95 | -0.517 | 0.61 | 0.23 |

Note. * Significant at alpha = .0055 level. $BF_{10}$ values below 0.33: moderate evidence for $H_0$; between 0.33 and 1: Weak evidence for $H_0$; between 1 and 3: Weak evidence for $H_1$; between 3 and 10: Moderate evidence for $H_1$; between 10 and 30: Strong evidence for $H_1$; between 30 and 100: Very strong evidence for $H_1$.

**Table 10. Effects of relatedness at different ROIs across different distractor conditions during the time window of -500 to -400 ms pre-response onset.**

| Time window | Distractor Type | Anteriority | df | Left | | | Middle | | | Right | | |
|---|---|---|---|---|---|---|---|---|---|---|---|---|
| | | | | t | p | $BF_{10}$ | t | p | $BF_{10}$ | t | p | $BF_{10}$ |
| -500 to -400 ms | Tonal Syllable | Anterior | 25 | 5.679 | < .001* | 3190.32 | 4.011 | < .001* | 65.54 | 3.895 | < .001* | 50.33 |
| | | Posterior | 25 | -4.157 | < .001* | 91.51 | -5.898 | < .001* | 5300.75 | -3.757 | < .001* | 36.93 |
| | Atonal Syllable | Anterior | 25 | 5.794 | < .001* | 4165.95 | 3.081 | 0.005 | 8.55 | 1.266 | 0.217 | 0.42 |
| | | Posterior | 25 | -1.502 | 0.146 | 0.56 | -4.527 | < .001* | 215.18 | -4.327 | < .001* | 135.51 |
| | Tone alone | Anterior | 25 | 1.243 | 0.225 | 0.41 | -1.575 | 0.128 | 0.62 | 1.009 | 0.322 | 0.33 |
| | | Posterior | 25 | 0.139 | 0.891 | 0.21 | 0.337 | 0.739 | 0.22 | -0.506 | 0.617 | 0.23 |

*Note.* * Significant at alpha = .0028 level. $BF_{10}$ values below 0.33: Moderate evidence for $H_0$; between 0.33 and 1: Weak evidence for $H_0$; between 1 and 3: Weak evidence for $H_1$; between 3 and 10: Moderate evidence for $H_1$; between 10 and 30: Strong evidence for $H_1$; between 30 and 100: Very strong evidence for $H_1$; Beyond 100: Extreme evidence for $H_1$.

evidence obtained in general favor the $H_0$ (i.e., with the $BF_{10}$ values < 1), with the only exception in the middle ROIs condition, where a $BF_{10}$ value of 1.95 was obtained, showing weak or inconclusive evidence in favor of $H_1$. Taken together, for the time window of -300 to -200 ms pre-response onset, the results of the above classical and Bayesian analyses both consistently indicated a robust left-oriented effect of Relatedness in the Atonal Syllable related condition. No other clear and comparable effects were found in the other two distractor conditions, which may account for the significant Relatedness x Type x Laterality interaction in this time window.

Note that a significant Relatedness x Anteriority x Laterality three-way interaction was also observed in the -500 to -400 ms time window, suggesting that the effect of Relatedness was not the same across the six ROIs. To examine if the same pattern existed in all three distractor type conditions (specifically, whether a significant Relatedness effect was also found in the Tone alone related condition), further *t*-tests were performed to investigate the simple effect of Relatedness in each ROI as a function of distractor types. Significant effects of Relatedness were found in both Tonal Syllable related and Atonal Syllable related conditions in various ROIs. Critically, null effects of Relatedness were found in the Tone alone related condition across different ROIs (*t*s < 1.6, *p*s > .12). Table 10 shows the results of the follow-up comparisons.

The effects of Relatedness were similarly examined using the Bayesian approach and the corresponding $BF_{10}$ values are shown in Table 10. Evidence in favor of $H_1$ (with the $BF_{10}$ values ranging from 36.93 [very strong evidence] to 5300.75 [extreme evidence]) was found in the Tonal Syllable related condition in all six ROIs. Evidence in favor of $H_1$ (with the $BF_{10}$ values ranging from 8.55 [moderate evidence] to 4165.95 [extreme evidence]) was also found in the Atonal Syllable related condition but in four ROIs only. Weak evidence in favor of $H_0$ was found in the right anterior and the left posterior sites in the Atonal Syllable condition. For the Tone alone related condition, however, the $BF_{10}$ values were all smaller than 1, indicating either moderate or weak evidence in favor of $H_0$.

### 3.3 Correlation analyses

In addition to the above behavioural and ERP analyses, attempts were made to investigate the relationship between the observed behavioural (priming) and ERP effects using correlational analyses. To this end, a number of difference scores capturing the difference between related and unrelated conditions in terms of their mean response time or mean ERP amplitude were

generated. We focused on those conditions where a significant behavioural or ERP effect was observed. For the behavioural data, significant priming effects were observed in the Tonal Syllable and the Atonal Syllable conditions (i.e., the RT in the related condition was significantly faster than that in the unrelated control condition). Two sets of RT difference scores (one from the Tonal Syllable condition and one from the Atonal Syllable condition) were therefore generated by subtracting the mean RT in the related condition from the mean RT of the corresponding unrelated control condition for each participant. For the present ERP data, significant ERP effects (i.e., the mean ERP amplitude of the related condition differed significantly from the unrelated control condition) were observed in the Tonal Syllable and Atonal Syllable conditions in various ROIs and in various stimulus-locked and response-locked time windows. Similar to the way of how the behavioural RT difference score was calculated, a similar approach was applied to calculate the difference scores in ERP (i.e., by subtracting the mean ERP amplitude of the related condition from that of the unrelated control condition for each participant). The ERP amplitude difference scores were derived from the time windows and ROIs of where a significant ERP effect was detected.

For instance, as reported in the above, a significant ERP effect was found in the Tonal Syllable condition in the 400–500 ms post-stimulus time window, and in the left ROIs. An ERP amplitude difference score was therefore generated for each participant by subtracting the mean ERP amplitude of the Tonal Syllable related condition from the corresponding unrelated control condition within the same 400–500 ms post-stimulus time window and left ROIs condition. The correlation between this ERP amplitude difference score and the corresponding RT difference score from the Tonal Syllable condition was then examined to see if there was any association between the two. Likewise, the same approach was applied to all other time windows and ROIs of where a significant ERP effect was observed.

The results of the correlation analyses are summarized in Table 11. Notably, significant correlations between RT difference scores and ERP amplitude difference scores were found in both Tonal Syllable and Atonal Syllable conditions in the time windows of -500 to -400 ms and -200 to -100 ms pre-response onset. No other correlations reached significance at $p$ = .05 level.

**Table 11. Correlations between RT difference scores and ERP amplitude difference scores in the Tonal Syllable related and the Atonal Syllable related conditions across different time windows and ROIs.**

| | | | Tonal Syllable | | Atonal Syllable | |
|---|---|---|---|---|---|---|
| | | N | Pearson's *r* | *p* | Pearson's *r* | *p* |
| Stimulus locked analysis | | | | | | |
| 400 to 500 ms | Left ROIs | 28 | 0.129 | 0.514 | 0.322 | 0.095 |
| Response locked analysis | | | | | | |
| -500 to -400 ms | Anterior ROIs | 26 | -0.415 | 0.035* | -0.516 | 0.007* |
| | Posterior ROIs | 26 | 0.484 | 0.012* | 0.583 | 0.002* |
| -400 to -300 ms | Anterior ROIs | 26 | -0.168 | 0.412 | N.A. | N.A. |
| | Posterior ROIs | 26 | 0.126 | 0.538 | N.A. | N.A. |
| -300 to -200 ms | Left ROIs | 26 | N.A. | N.A. | 0.218 | 0.285 |
| -200 to -100 ms | Anterior ROIs | 26 | 0.392 | 0.047* | 0.451 | 0.021* |
| | Posterior ROIs | 26 | -0.358 | 0.072 | -0.474 | 0.014* |
| -100 to 0 ms | Anterior ROIs | 26 | 0.062 | 0.765 | N.A. | N.A. |
| | Posterior ROIs | 26 | -0.017 | 0.933 | N.A. | N.A. |

*Note.* * Significant at alpha = .05 level. N.A. means that no correlation analysis was performed for that particular condition as no significant ERP effect (i.e., the mean ERP amplitudes in the related and unrelated control conditions did not differ significantly from each other) was observed in that condition.

## 4. Discussion

### 4.1 Summary of the results

A picture-word interference experiment with concurrent ERP recording was conducted to investigate the time course of tonal and syllabic encoding in Cantonese spoken word production. Consistent with the findings from previous relevant behavioural studies, participants' naming latencies were faster, relative to an unrelated control, in the Tonal Syllable related and Atonal Syllable related conditions but null effect was found in the Tone alone related condition. Furthermore, the Tonal Syllable priming effect was significantly larger than that of the Atonal Syllable priming effect, indicating that although the effect of Tone alone is weak, it can possibly be detected upon concurrent segmental priming. In addition, significant effects of Relatedness on the ERPs were found in both Tonal Syllable related and Atonal Syllable related conditions in the 400–500 ms time window post-stimulus. The effects were similarly left-oriented. The similarity in timing and pattern of the ERP effects (by stimulus-locked analysis) across these two phonological conditions indicate that the effects may arise from the priming of the atonal syllable component (since both Tonal Syllable related and Atonal Syllable related distractors shared the same atonal syllable unit with their corresponding targets). Furthermore, significant effects of Relatedness on the ERPs were also found in both Tonal Syllable related and Atonal Syllable related conditions in several 100-ms time windows prior to response onset. Notably, the two phonological conditions showed distinct ERP effects in the time windows of 400-to-300 ms and 100-to-0 ms pre-response onset, indicating an additional effect of tone (over and beyond the atonal syllable unit) at a relatively late stage of speech production. However, null effect on ERPs was found in the Tone alone related condition in both stimulus-locked and response-locked analyses, consistent with the corresponding null behavioural result.

### 4.2 Null effect of the tone alone related condition

The null effect of the Tone alone related condition stands in marked contrast to the significant effects observed in the Atonal Syllable related condition, consistent with the results of some previous Chinese speech error studies which suggested that tone is represented or processed differently from segmental units [15]. Interestingly, in almost all published Chinese speech production studies where the tonal dimension was manipulated, the effect of tone alone (e.g., the prime and target shared the same tone only) on naming response has rarely been found. Facilitation effects on naming latency have repeatedly been found with different speech production paradigms when the atonal syllable or sub-syllabic components (e.g., syllable body or rhyme) of the target were primed or known in advance [17,21–23,52,53]. A similar situation occurred concerning the effect of onset consonant in Chinese language production where a significant effect of onset alone was rarely observed (but see [54], for an exception). Although behavioural effects associated with the onset consonant alone were rarely seen in Chinese, significant ERP effects associated with the onset component have been reported in the literature [32,33]. The present study was therefore conducted to examine the effect of tone alone using concurrent ERP recordings, yet null effects on the ERPs were observed in the Tone alone related condition in both stimulus-locked and response-locked analyses. The null effects of the Tone alone related condition were in line with the results of Bayesian analyses where weak to moderate evidence supporting the null hypothesis ($H_0$) was found in the Tone alone related condition in almost all time windows (with the only exception in the -300 to -200 ms pre-response time window where weak or inconclusive evidence supporting $H_1$ was found in the middle ROIs), indicating that a lack of task sensitivity was probably not the major reason for the null behevioural and ERP effects. The contrasting findings between tone and segmental

units (e.g., atonal syllable and individual segments) suggest that tone is probably represented or processed differently from segmental information in Chinese language production. A similar conclusion has been reached in a recent tongue twister study in Mandarin [55]. These authors found non-negligible number of tonal errors in a Mandarin tongue twister experiment. Importantly, the tonal errors observed showed different patterns from the segmental errors, indicating that tone is implicated in Chinese speech production but is processed differently from the segmental units (see also [56], for distinctive roles of tonal and segmental units in Mandarin speech perception). Findings from neuroimaging studies also showed distinct neural substrates and activation patterns responsible for tonal and segmental processing in Chinese speech processing [57,58].

The notion that tone and segmental information involve different encoding mechanisms is consistent with the models of Chinese speech production proposed by [19,20]. Both models assume that in Chinese phonological encoding, the atonal syllable unit is retrieved first by selecting from an inventory of possible atonal syllables in the language. Dissimilar to the segmental content, however, the tone is mapped onto a frame-like structure which specifies the supra-segmental property of the target. The different treatments for tone and segmental content in the early stage of phonological encoding may account for their different proneness to show significant priming effects under experimental manipulations. In contrast, the distinctive findings between tone and segmental units are not consistent with the early encoding view (e.g., [13]) which assumes that tones and phonemes are similarly processed and selected at the early stage of phonological encoding. According to this latter view, the effects of tone and segmental units (i.e., the atonal syllable in this study) should both appear at an early ERP time window. However, the present ERP results do not provide evidence supporting an early effect of tone similar to that of the segmental information.

### 4.3 Atonal Syllable as the proximate unit in Cantonese phonological encoding

Another interesting finding of the present study relates to the processing of atonal syllable in Cantonese. A significant effect of Relatedness was found in the Atonal Syllable related condition in the 400–500 ms time window post-stimulus onset, which might reflect the time of which the atonal syllable was retrieved in Cantonese phonological encoding. Note that a similar left-oriented ERP effect was also observed in the Tonal Syllable related condition in the same time window, indicating that the effect was associated with segmental priming (since both Tonal Syllable and Atonal Syllable distractors shared the same segmental contents with the targets). In a comprehensive analysis with a large number of language production studies, [59] (see also [60]) suggested that in a typical picture naming event, the first 200 ms post-stimulus presentation is associated with visual processing and concept activation, then followed by approximately 75 ms of lexical selection. Phonological encoding starts at approximately 275 ms and lasts until 455 ms post-stimulus, and the last 145 ms is devoted to phonetic encoding and articulation. Previous behavioural and recent ERP studies suggested that the proximate unit (i.e., the first retrievable phonological unit following lexical selection) in Chinese phonological encoding is atonal syllable [17,19,35]. The ERP effects presently observed in the 400-500-ms time window post-stimulus might therefore suggest that phonological encoding in Cantonese begins at around 400 ms post-stimulus, which is apparently later than the estimation made by [59] (with a difference of 125 ms in their onset). However, [59] assumed a mean picture naming latency of 600 ms without interference from distractors. The overall mean naming latency of the present study is 740 ms which is considerably (140 ms) longer than the mean naming latency assumed by [59]. Together with the presentation of an auditory

distractor in each trial, it is conceivable that the absolute timing of the ERP effect associated with atonal syllable priming appeared in a time window which is later than the estimated time course proposed by [59]. In addition, it has been suggested that the time taken for lexical selection accounts for a considerably large portion of variance in naming latency differences, and hence one way to align the time frame of [59] with the present data is to re-adjust the estimated duration for lexical selection, and assume that the time difference between the two mean naming latencies (i.e., 140 ms) is attributed to a lengthened lexical selection process (i.e., from the originally proposed 75 ms to 215 ms). Accordingly, in the context of the present experiment, phonological encoding started at approximately 415 ms and lasted until 595 ms post-stimulus. The observed ERP effects associated with atonal syllable in the 400–500 ms time window post-stimulus are consistent with the notion that atonal syllable is the proximate unit in Cantonese phonological encoding.

## 4.4 Time course of tonal and syllabic encoding in Cantonese

The absolute timing of the ERP effects might be subject to the influence of a number of factors, what is more informative perhaps is the relative timing of the different ERP effects arising from different conditions of the same experiment. Although similar ERP effects were observed in the Tonal Syllable related and Atonal Syllable related conditions in the stimulus-locked analysis, the ERP effects in these two phonological conditions diverged from each other in the response-locked analysis. Since stimulus-locked analysis is presumably more sensitive to the early stage of processes associated with stimulus processing whereas response-locked analysis is more sensitive to the late stage of processes in relation to the overt naming response [38], the divergent ERP effects observed between Tonal Syllable related and Atonal Syllable related conditions in the response-locked analysis indicate a tone-related effect at a relatively late stage of speech preparation. This is because the critical difference between Tonal Syllable related and Atonal Syllable related distractors is their tonal overlap with the target, only the Tonal Syllable related distractors shared the same tone with the target but not the Atonal Syllable related distractors. Therefore, the difference in their ERP effects should be attributed to the additional effect of tone. The larger behvioural priming effect (i.e., the over additive effect of tone) in the Tonal Syllable related condition than the Atonal Syllable related condition is consistent with the notion that tone is implicated in Cantonese speech preparation and its effect can be shown upon segmental priming. The present ERP data indicate that the over additive effect of tone has a relatively late locus. The present results suggest that in Cantonese phonological encoding, the atonal syllable is retrieved first, followed by a process of atonal syllable to tone association. The results of response-locked analysis are also consistent with this notion. In the earliest response-locked time window (500-to-400 ms pre-response), comparable ERP effects were observed across Tonal Syllable related and Atonal Syllable related conditions. The effects of the two phonological conditions started to diverge from each other from the time window of 400-to-300 ms pre-response during which a significant effect of relatedness was found only in the Tonal Syllable related condition, consistent with the notion that atonal syllable retrieval precedes the syllable-to-tone association stage. These results are consistent with the late encoding views which assume that in Chinese phonological encoding, the atonal syllable is retrieved first and then associate with the tone with the tonal property specified at a later stage of phonological encoding [19,20]. Since the syllable-to-tone association stage requires both tone and segmental content to be specified, only the Tonal Syllable related (but not the other two distractor conditions) can possibly exert priming at this stage. The significant effect of the Tonal Syllable related condition, together with the null effects of the other phonological conditions, in the time window of -400 to -300 ms pre-response are consistent with this interpretation.

Note that the present ERP effects observed with the response-locked analysis appear to occur in two different phases. Specifically, significant ERP effects were observed in the Tonal Syllable condition in the time windows of -500 to -300 ms and -200 to 0 ms pre-response onset. Likewise, significant ERP effects were observed in the Atonal Syllable condition in the time windows of -500 to -400 ms and -300 to -100 ms pre-response onset. A tentative account for this bi-phasic pattern of results might be due to the reason that the effects were arising from multiple processing stages, with phonological encoding first, followed by phonetic encoding and articulation. This proposed account is also consistent with the processing stages in word production as assumed by [59]. Conversely, results of the correlation analyses also showed that the ERP effects observed in the Tonal Syllable related and Atonal Syllable related conditions in the time windows of -500 to -400 ms and -200 to -100 ms correlated significantly with the behavioural priming effects, indicating that the phonological priming effects presently observed using the PWI task might have more than one locus. It is conceivable that the Tonal Syllable related distractors might affect both phonological encoding and the subsequent phonetic encoding and articulation processes since the target and distractor share also the same corresponding articulatory gestures in their word-initial syllable position. For the Atonal Syllable related distractors, although they are different from the target in their tones, they do share all the same segmental content with the first syllable of the target, they may therefore capable to exert certain degree of priming even at the phonetic encoding or articulatory processes. Nevertheless, the present data do not provide clear evidence to verify these possibilities. Future studies manipulating the syllable frequency of the target (e.g., [61]) might be a viable way to verify the nature of the bi-phasic pre-response ERP effects observed in the present study.

The present study is the first to investigate the time course of tonal and syllabic encoding in overt Chinese speech production using the PWI task with concurrent ERP recording. Stimulus-locked and response-locked analyses were conducted to reveal the early and the late processes in spoken word planning respectively. Significant and similar ERP effects were observed in the Tonal Syllable related and the Atonal Syllable related conditions in stimulus-locked analysis, as well as in the early time window of the response-locked analysis. Distinct effects between these two phonological conditions were not observed until -400 to -300 ms pre-response onset in the response-locked analysis. These results are consistent with the late encoding view, suggesting that in Chinese phonological encoding, the atonal syllable is retrieved first, followed by the tone-to-syllable association process at a later stage of phonological encoding [19,20].

## Supporting information

**S1 Appendix. List of targets and distractors.**
(DOCX)

## Author Contributions

**Conceptualization:** Andus Wing-Kuen Wong, Yiu-Kei Tsang, Hsuan-Chih Chen.

**Data curation:** Ho-Ching Chiu.

**Formal analysis:** Andus Wing-Kuen Wong, Ho-Ching Chiu.

**Funding acquisition:** Andus Wing-Kuen Wong.

**Investigation:** Andus Wing-Kuen Wong, Ho-Ching Chiu.

**Methodology:** Andus Wing-Kuen Wong, Ho-Ching Chiu.

**Project administration:** Andus Wing-Kuen Wong, Ho-Ching Chiu.

**Resources:** Andus Wing-Kuen Wong, Yiu-Kei Tsang.

**Software:** Yiu-Kei Tsang.

**Supervision:** Andus Wing-Kuen Wong, Yiu-Kei Tsang, Hsuan-Chih Chen.

**Writing – original draft:** Andus Wing-Kuen Wong, Ho-Ching Chiu.

**Writing – review & editing:** Andus Wing-Kuen Wong, Ho-Ching Chiu.

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
