## [Decision Letter · Decision Letter 0]

5 Jul 2023

PONE-D-23-08190Tonal and syllabic encoding in overt Cantonese Chinese speech production: An ERP studyPLOS ONE

Dear Dr. Chiu,

Thank you for submitting your manuscript to PLOS ONE. After careful consideration, we feel that it has merit but does not fully meet PLOS ONE’s publication criteria as it currently stands. Therefore, we invite you to submit a revised version of the manuscript that addresses the points raised during the review process.

We look forward to receiving your revised manuscript.

Kind regards,

Caicai Zhang

Academic Editor

PLOS ONE

Journal Requirements:

"The work described in this paper was fully/partially supported by research grants from the Research Grants Council of the Hong Kong Special Administrative Region, China (Project No. CityU 21402514 and CityU 11673316).

"We note that you have provided additional information within the Acknowledgements Section that is not currently declared in your Funding Statement. Please note that funding information should not appear in the Acknowledgments section or other areas of your manuscript. We will only publish funding information present in the Funding Statement section of the online submission form.

"The work described in this paper was fully/partially supported by research grants from the Research Grants Council of the Hong Kong Special Administrative Region, China (Project No. CityU 21402514 and CityU 11673316). The funders had no role in study design, data collection and analysis, decision to publish, or preparation of the manuscript."

Reviewers' comments:

Reviewer's Responses to Questions

**Comments to the Author**

1. Is the manuscript technically sound, and do the data support the conclusions?

Reviewer #1: Yes

Reviewer #2: Partly

2. Has the statistical analysis been performed appropriately and rigorously? 

Reviewer #1: Yes

Reviewer #2: Yes

3. Have the authors made all data underlying the findings in their manuscript fully available?

Reviewer #1: Yes

Reviewer #2: Yes

4. Is the manuscript presented in an intelligible fashion and written in standard English?

Reviewer #1: Yes

Reviewer #2: Yes

5. Review Comments to the Author

Reviewer #1: This study investigated how syllables and lexical tones are encoded in Cantonese speech production using a picture-word interference task with concurrently recoding electrophysiological signals. Findings suggest that in Cantonese speech production, the atonal syllable of the target is retrieved and then associated with the lexical tone. The manuscript is well written. However, I have some major concerns.

1. In the introduction, the authors cited the additive factors method proposed by Sternberg (1969), however, they did not manipulate factors of tone and segments with a factorial design. I think it is questionable to apply the additive factors method in the present design. On the other hand, this logic and paradigm have been widely questioned by many researchers (Koch, Poljac, Müller, & Kiesel, 2018). Therefore, it should be discussed and justified whether the logic is usable in the present study.

2. The major argument is based on the null effect of tonal alone related condition. Please include details of a priori power analysis, a sensitivity analysis, or some other way to make a case that your experiments are sufficiently sensitive or precise enough for your goals.

3. I suggest to report Bayesian factor analyses for all statistical results.

4. For spoken word production, how to align the time course of processing found in stimulus locked and response locked analyses?

5. It is difficult to understand why the contrasting findings between tone and segmental units reflect that tone is represented of processed differently from segments. I suggest discuss what are the differences between them.

Reviewer #2: This interesting paper tackles the less-studied question: the time course of the lexical tones in the Chinese language and the whole study makes contributions to the field of Chinese word production. The overall design is valid and the results are clearly presented. But I have some questions about the paper. Please see the attached comments.

6. PLOS authors have the option to publish the peer review history of their article (what does this mean?). If published, this will include your full peer review and any attached files.

Reviewer #1: No

Reviewer #2: No

---

## [Author Response · Author response to Decision Letter 0]

5 Oct 2023

Re: Revised version of the manuscript titled “Tonal and syllabic encoding in overt Cantonese Chinese speech production: An ERP study” 

We are deeply grateful for the valuable and constructive comments from the reviewers in the previous round of review, based on which substantial revisions to the manuscript have been made. Details of our response to the reviewers’ comments are listed in the following.

Reviewers' comments:

Reviewer #1: This study investigated how syllables and lexical tones are encoded in Cantonese speech production using a picture-word interference task with concurrently recoding electrophysiological signals. Findings suggest that in Cantonese speech production, the atonal syllable of the target is retrieved and then associated with the lexical tone. The manuscript is well written. However, I have some major concerns.

Response: We thank Reviewer 1 for the positive evaluations of this work.

1. In the introduction, the authors cited the additive factors method proposed by Sternberg (1969), however, they did not manipulate factors of tone and segments with a factorial design. I think it is questionable to apply the additive factors method in the present design. On the other hand, this logic and paradigm have been widely questioned by many researchers (Koch, Poljac, Müller, & Kiesel, 2018). Therefore, it should be discussed and justified whether the logic is usable in the present study.

Response: Thank you for pointing this out. We agree that the present design does not entirely follow a 2 x 2 factorial design and hence it might be questionable if the additive factors logic can readily be applied to this case. Instead of using an independent set of distractor materials to be the baseline (i.e., phonologically unrelated condition) for comparison to fit into and complete the 2 x 2 design, the same three sets of phonologically related distractors (i.e., Tonal Syllable related, Atonal Syllable related, and Tone alone related distractors) were used as the baselines by re-grouping the target-distractor pairs. This way, the same set of picture and distractor materials was used in each phonological condition, and the only difference between related and unrelated conditions was the target-distractor relationship. A within-subjects and within-items design was then achieved. We felt this could best minimize any potential confound associated with the items’ properties, and hence the unrelated baseline conditions were constructed in such a manner. Nevertheless, we agree with Reviewer 1 and have revised the part mentioning about Sternberg’s (1969) additive factors logic. In the revised version, the present design is now motivated by the different views (e.g., early or late tonal encoding) in relation to tonal and syllable encoding as assumed by the different theories of Chinese speech production. Corresponding revisions have been made to the Introduction of the revised version.

2. The major argument is based on the null effect of tonal alone related condition. Please include details of a priori power analysis, a sensitivity analysis, or some other way to make a case that your experiments are sufficiently sensitive or precise enough for your goals.

Response: The sample size of the present study is comparable to the previous related ERP (e.g., Qu et al., 2012, 2020) and behavioural (Wang et al., 2019) studies where a significant effect of a single phoneme on Chinese speech planning has been reported. We therefore believe that the current sample size should be sufficient enough to detect the effect of tone alone, if any. Furthermore, additional analyses using the Bayesian approach have been conducted and reported in the revised version. The results of Bayesian analyses showed evidence in support of the H0 (i.e., evidence supporting the absence of an effect) in the Tone alone related condition in almost all ERP time windows (with the only one exception showing an inconclusive evidence supporting H1). Consequently, regarding the Tone alone related condition, the present ERP data (using the PWI task) suggest that there is evidence in favor of the null effect of tone alone instead of an absence of evidence for any effect.

3. I suggest to report Bayesian factor analyses for all statistical results.

Response: We thank the Reviewer for this valuable suggestion. It has been done accordingly. The additional results are largely consistent with the results originally reported. The overall picture has become even more clear with the inclusion of Bayesian factor analyses in the revised manuscript. 

4. For spoken word production, how to align the time course of processing found in stimulus locked and response locked analyses?

Response: We thank the Reviewer for pointing this out. This is a very sensible question. Yet, we are inclined to believe that the present data do not provide us with enough grounds to fully address this concern, future studies (especially studies on the methodological issues related to stimulus-locked and response-locked analyses in particular) might be warrant. As argued by Riès et al. (2013), early stimulus-evoked activities are steeper in their slopes and more transient (as they are well averaged to the stimulus) than the later decision/ selection-evoked activities, it might still be an open issue on how to map the absolute timing between the later part of stimulus-locked analysis and the earlier part of response-locked analysis. Nevertheless, we have adopted the general principles from the previous studies (e.g., Jeong et al., 2019; Laganaro, 2014; Riès et al., 2013) in that stimulus-locked and response-locked analyses are sensitive to the early and late stages of processing respectively. As discussed in the revised manuscript, the effect of atonal syllable priming has been used as a reference to infer the time of when phonological encoding begins (since the bulk of evidence in the literature suggests that the atonal syllable is the first retrievable phonological unit in Chinese phonological planning). With this reference, the relative time course of tonal encoding (e.g., the joint effect of tonal and syllabic priming) was possibly examined in this study. 

5. It is difficult to understand why the contrasting findings between tone and segmental units reflect that tone is represented of processed differently from segments. I suggest discuss what are the differences between them.

Response: Revised accordingly. Elaborations on the potential differences between tonal and segmental encoding (in the way how they are processed), as postulated by different theories of Chinese speech production, have been included in the revised manuscript. 

Reviewer #2: This interesting paper tackles the less-studied question: the time course of the lexical tones in the Chinese language and the whole study makes contributions to the field of Chinese word production. The overall design is valid and the results are clearly presented. But I have some questions about the paper. Please see the attached comments.

Response: We thank Reviewer 2 for the positive evaluations of this work.

1. In the last paragraph of the Introduction, the authors mentioned that "if tone and atonal syllable are processed together and affecting the same processing stage, the effects of of tone and atonal syllable relatedness on the ERPs should be non-additive". The logic underlying this claim is not very clear. It will be better for the authors to elaborate on this. Moreover, it will be also better to discuss how the current EEG results relate to these hypotheses again in the discussion section.

Response: Thank you for pointing this out. We agree that the present design does not entirely follow a 2 x 2 factorial design and hence it might be questionable if the additive factors logic can readily be applied to this work. The corresponding discussions have been removed in the revised manuscript. In the revised Introduction, the present study is motivated by the different views (e.g., early or late tonal encoding) in relation to tonal and syllable encoding as assumed by the different theories of Chinese speech production. In addition, elaborated discussions have been included in the Discussion part to relate the present results to the different hypotheses mentioned in Introduction. 

2. The authors mentioned Alderete et al. (2019), who also proposed a new model for the tonal encoding in Chinese production. It will be more insightful for the authors to also review their model in this paper and make some comparisions with the previous models proposed by O' Seaghdha et al. (2010) and Roelofs (2015) in the literature review. It will be also better to discuss how the current ERP results could disentangle these different models in the discussion session.

Response: Revised accordingly. Further reviews on Alderete and colleagues’ (2019) model and comparisons between this and other models of Chinese speech production have been provided in the revised manuscript. In addition, elaborated discussions have been added to the Discussion section to discuss how the present ERP data can possibly differentiate the various hypotheses. 

3. The authors mentioned one participant was left-handed. Did this author include this participant in the EEG data analysis? Have you checked whether this participant influenced the overall result?

Response: The left-handed participant was included in the stimulus-locked analysis but was not in the response-locked analysis. Nevertheless, a similar pattern of result was observed whether or not this participant was included in the ERP data analysis, the conclusions reached in this study would not be affected by this single case. 

4. For the distractors, the authors used disyllabic items. Why did the authors not use monosyllabic distractors? Any considerations for this? Have the authors checked that the second syllable of the distractors was not phonologically related to the paired target item?

Response: Thank you for this suggestion. We had considered this possibility before. It was only that if monosyllabic distractors were used, the Tonal Syllable related distractors will then become identical (not merely homophonous) to the first syllable/character/morpheme of the target. The target and tonal syllable related distractors will then share many aspects of similarity (e.g., semantic, morphological, phonological, and orthographic) apart from the phonological one. The situation would be different if disyllabic distractors were used, where the second syllable of the distractor could provide the necessary context guiding the interpretation of the first syllable (which was only homophonous but not identical to the target). Disyllabic distractors were therefore adopted in the present study. Cautions have been taken to ensure that the second syllable of the distractor did not share the same atonal syllable with the targets. 

5. The authors mentioned that the three types of distractors were matched in the syllable frequency and number of homophone. Did the authors match these two variables for both syllables of the distractors?

Response: Thank you for pointing this out. Clarifications have been made in the revised manuscript (section 2.2 Materials and apparatus). 

6. It will be better to provide some details about the preparation of the auditory distractors (were the items pronounced by female or male speakers? What about the average duration of the audios (Was it matched between the three types of distractors)?

Response: Provided accordingly. Please see the revised section 2.2. The mean duration of the auditory distractors was matched across the three distractor type conditions. 

7. Have the authors also checked that the second syllable of the Cantonese target items were not phonologically related to the first syllable of the items, as the distractors could also impact the phonological encoding of the second syllable of the items and influenced the ERP results? It will be better for the authors to provide a full list of the stimuli in the appendix.

Response: Cautions have been taken to ensure that the two syllables of the distractor did not overlap in their atonal syllable. A full list of the stimuli is included in the Appendix of the revised manuscript. 

8. Inspection of the stimulus-locked and response-locked ERPs showed some inconsistencies. In principle, there should be a temporal overlap between the later portion of the stimulus-locked ERP and the early portion of response-locked ERP, which typically would show some converging patterns. However, the later portion of the stimulus-locked ERP showed opposite ERP patterns to those in the early portion of the response-locked ERP. For example, Figure 2A showed that the tonal syllable related condition showed more negativity in the left anterior region than the unrelated condition in the 400-500ms post stimulus onset, but Figure 3A showed the opposite polarity for the tonal syllable related condition in the left anterior region in the -500- -300ms before the speech onset. I noticed the authors had different numbers of participants and different trials for the stimulus-locked and the response-locked ERPs, as well as using different baselines for correction. I think that the authors may need to follow Jeong et al. (2021) and use the same trials and participants and the same baseline for both stimulus-locked and response-locked ERP analysis to see whether such divergent patterns in the paper still existed. 

Jeong, H., van den Hoven, E., Madec, S., & Bürki, A. (2021). Behavioral and Brain Responses Highlight the Role of Usage in the Preparation of Multiword Utterances for Production. Journal of Cognitive Neuroscience, 1–34. https://doi.org/10.1162/jocn_a_01757

Response: We thank Reviewer 2 for this valuable comment. We agree that the different polarities of the effects mentioned above might be related to the differences between stimulus-locked and response-locked analyses presently conducted. Yet, we also noticed that it might not be very uncommon for researchers to adopt different baselines for stimulus-locked and response-locked analyses (e.g., Riès et al., 2013). In our response-locked analysis, the 200-ms period between 2000 ms and 1800 ms pre-response onset was taken as the baseline, to make sure that the baseline period would fall entirely within the time of when the fixation point was presented which was the same for all trials and all conditions included in the analysis. Another merit of this approach is that the time lag between the baseline period and the time window of interest (i.e., the 500-ms pre-response onset) could be kept constant across different conditions and trials. This time lag, however, would be varied across conditions (and trials) if a fixed baseline period (say for instance, the 200-ms pre-stimulus onset) was chosen for the response-locked analysis (since the response onset varies across conditions and trials).

Nevertheless, we have adopted the general principles from the previous related studies (e.g., Jeong et al., 2019; Laganaro, 2014; Riès et al., 2013) in that stimulus-locked and response-locked analyses are sensitive to the early and late stages of processing respectively. As discussed in the revised manuscript, the effect of atonal syllable priming has been used as a reference to infer the time of when phonological encoding begins (since the bulk of evidence in the literature suggests that the atonal syllable is the first retrievable phonological unit in Chinese phonological planning). With this reference, the relative time course of tonal encoding (e.g., the joint effect of tonal and syllabic priming) was possibly examined in this study. Although it might still be an open issue regarding the different polarities of the effects mentioned above, the current ERP results are clear enough to inform the various views on the relative time course of tonal (early or late encoding) and syllabic encoding in Chinese speech production. 

9. The authors mentioned that there are six blocks with each block containing the six distractor conditions. Did the authors counterbalance or randomize the order of the blocks across participants?

Response: Clarifications have been provided accordingly (please see revised section 2.3 Design and procedure).

10. It seemed that the authors used the voice key automatic detection for the response latencies measurement. Have the authors checked the reliability and accuracy for the detected response latencies using the recorded audios? 

Response: We thank Reviewer 2 for this valuable suggestion. This was not included in our original research plan and proposal so no audio recording of the participants’ responses was taken for this purpose. This is a limitation of the present study. We totally agree that it would be very useful to have this included and will take this into consideration in our future studies.

11. In the behavioral analysis, the authors excluded the reaction times shorter and longer than 3 SDs, but in the EEG analysis, the authors excluded those trials less than 500 ms and longer than 1500 ms. Why did the authors choose to use different outlier exclusion threshold for the behavioral and EEG data analysis respectively？

Response: Excluding behavioural trials with reaction times beyond 3 SDs was a practice generally used in the past behavioural studies. In our ERP analysis, trials with RT less than 500 ms or longer than 1500 ms were discarded. Trials with RT less than 500 ms were excluded so as to make sure that the response onset would not occur within the first 500 ms post-stimulus onset where the stimulus-locked analysis was conducted, to minimize the impacts of motor related artefacts. Trials with RT longer than 1500 ms were also excluded because for those trials, the baseline period (i.e., the 2000 ms to 1800 ms pre-response onset) would fall out of the period of when the fixation point was presented on the screen. Nevertheless, the difference in exclusion criterion was due mainly to practical methodological reasons, and there was a considerable degree of overlap between these different criterion (i.e., trials with RT beyond 3 SDs were also the trials with RT lower than 500 ms or longer than 1500 ms). 

12. For the last two paragraphs of 3.2.2, the authors need to provide additional tables to show the results of the further t-test analyses for the Relatedness x Type x Laterality in the -300 to -200ms and the Relatedness x Anteriority x Laterality in the -500 to -400 ms.

Response: Provided accordingly.

13. It seemed that the authors observed a larger priming effect for the tonal syllable related distractors than the atonal syllable related distractors in the behavioral data, but it is unclear how this relates to the EEG results. It seemed that the authors did not observe any amplitude difference for the two types of distractors but only found that the two types of distractors showed distinct patterns in the -400 to -200 and -100 to 0 time window. How could we explain the relatedness between the behavioral data pattern and these EEG results? Why were there two separate time windows showing the different effects? What does this indicate? Some tentative discussion on this would be helpful.

Response: We thank Reviewer 2 for this valuable comment. Additional correlation analyses have been conducted and reported in the revised manuscript to provide supplemental evidence to show the relationship between the observed behavioural and ERP effects. In addition, further elaborated discussions, as well as a tentative account, for the finding that two separate time windows in response-locked analysis showing the different effects of the Tonal Syllable and Atonal Syllable conditions are included in the revised Discussion section. 

14. Table 3 The authors need to add the table header in the first row to denote the meaning of the numbers. 

Response: Revised accordingly.

15. The ERP figures are too blurry with low resolution. Clearer figures need to be provided.

Response: Revised accordingly.

16. other minor issues 

(a) Para. 2, Page 9 (Introduction section) "in a slips of the tongue study" -> "in a study on slips of the tongue"

Response: Revised accordingly.

(b) Para 2, Page 13 (last paragraph in the introduction) Line 5 "Tonal related condition" -> "Tonal syllable related condition" (Please check the whole paper for consistency)

Response: Revised accordingly.

(c) Page 15 (2.2 section) "incorrect response (1.6%)" and Page 16 (3.1) "Incorrect (1.58%)" Please keep the consistency of the number of digits.

Response: Revised accordingly.

---

## [Decision Letter · Decision Letter 1]

30 Oct 2023

PONE-D-23-08190R1Tonal and syllabic encoding in overt Cantonese Chinese speech production: An ERP studyPLOS ONE

Dear Dr. Chiu,

Thank you for submitting your manuscript to PLOS ONE. After careful consideration, we feel that it has merit but does not fully meet PLOS ONE’s publication criteria as it currently stands. Therefore, we invite you to submit a revised version of the manuscript that addresses the points raised during the review process.

We look forward to receiving your revised manuscript.

Kind regards,

Caicai Zhang

Academic Editor

PLOS ONE

Journal Requirements:

Additional Editor Comments (if provided):

Two reviewers have reviewed the revised manuscript and are content with the revision. Reviewer 2 raised two minor issues with regard to data checking and stimulus control. I encourage the authors to take into consideration these remaining suggestions and acknowledge them as limitations and future directions if necessary.

Reviewers' comments:

Reviewer's Responses to Questions

**Comments to the Author**

1. If the authors have adequately addressed your comments raised in a previous round of review and you feel that this manuscript is now acceptable for publication, you may indicate that here to bypass the “Comments to the Author” section, enter your conflict of interest statement in the “Confidential to Editor” section, and submit your "Accept" recommendation.

Reviewer #1: All comments have been addressed

Reviewer #2: All comments have been addressed

2. Is the manuscript technically sound, and do the data support the conclusions?

Reviewer #1: Yes

Reviewer #2: Yes

3. Has the statistical analysis been performed appropriately and rigorously? 

Reviewer #1: Yes

Reviewer #2: Yes

4. Have the authors made all data underlying the findings in their manuscript fully available?

Reviewer #1: Yes

Reviewer #2: Yes

5. Is the manuscript presented in an intelligible fashion and written in standard English?

Reviewer #1: Yes

Reviewer #2: Yes

6. Review Comments to the Author

Reviewer #1: Authors revised the manuscript according to my reviews. I think it is acceptable for publication in PLoS ONE.

Reviewer #2: The questions raised have been mostly addressed. However, there are still two issues for the authors.

First, the use of different baseline for stimulus-locked and response-locked ERP was justified by the authors' explanations. However, it will be better for the authors to check if the same participants and the same trials used for stimulus-locked and response-locked ERPs produced the similar results as reported in the current version, as this could ensure that the opposite polarity between the later portion of stimulus-locked ERP and the early portion of response-locked ERP was not just due to difference in participants and trials (given the intersubject and intertrial variability typically observed in ERP data).

Second, the stimulus selection/control in the current experiment was actually not very perfect. I noticed that some of the target items contained two different syllables with the same tone (e.g., fei1 kei1, 'plane'). Since the tone is the focus of the current study, it will be better to try to avoid such items in order to obtain a more reliable effect, as it is unclear whether the tone itself could potentially show some effect in ERP. Even though previous studies did not report the tonal overlap alone could produce priming effect in behavioral data (in Cantonese and Mandarin), there is some evidence that the tonal overlap alone may potentially cause an interfernce effect in Mandarin (see Chen et al., 2002). Moreover, the rime sharing (e.g., fei1 kei1, 'plane') may also introduce some unknown influence to the processing.

7. PLOS authors have the option to publish the peer review history of their article (what does this mean?). If published, this will include your full peer review and any attached files.

Reviewer #1: **Yes: **Qingfang Zhang

Reviewer #2: No

---

## [Author Response · Author response to Decision Letter 1]

17 Nov 2023

Re: 2nd revision of the manuscript titled “Tonal and syllabic encoding in overt Cantonese Chinese speech production: An ERP study” 

We are deeply grateful for the valuable and constructive comments from the reviewers in the last two rounds of review, based on which revisions to the manuscript have been made. Details of our response to the reviewers’ comments are listed in the following.

Reviewers' comments:

Reviewer #1: Authors revised the manuscript according to my reviews. I think it is acceptable for publication in PLoS ONE.

Response: 

We thank Reviewer 1 for the positive evaluation of this work.

Reviewer #2: The questions raised have been mostly addressed. However, there are still two issues for the authors.

First, the use of different baseline for stimulus-locked and response-locked ERP was justified by the authors' explanations. However, it will be better for the authors to check if the same participants and the same trials used for stimulus-locked and response-locked ERPs produced the similar results as reported in the current version, as this could ensure that the opposite polarity between the later portion of stimulus-locked ERP and the early portion of response-locked ERP was not just due to difference in participants and trials (given the intersubject and intertrial variability typically observed in ERP data).

Response:

We thank Reviewer 2 for the suggestion. We agree with Reviewer 2 that multiple factors may contribute to the different ERP waveform patterns between the end of the stimulus-locked time window and the beginning of the response-locked time window. To examine the potential influence of inter-subject variability, additional ERP analyses have been conducted with the data from the 24 participants which were previously included in both stimulus-locked and response-locked analyses. This way, the same group of participants (though with a reduced sample size) was included in the additional analysis for both stimulus-locked and response-locked analyses (using similar procedures as previously described). The results of the additional analyses were largely similar to the stimulus-locked and response-locked analysis results previously presented in the report (details are shown in the tables below). Specifically, significant ERP effects (i.e., significant differences between related and unrelated conditions in mean ERP amplitudes) were observed in both Tonal Syllable and Atonal Syllable distractor conditions in the time window of 400 to 500 ms post-stimulus onset (i.e., the end of the stimulus-locked time window), and the effects were confined to the anterior ROIs. Significant ERP effects were also observed in the Tonal Syllable and Atonal Syllable conditions in the response-locked analysis across various time windows, including the -500 to -400 ms pre-response onset (i.e., the beginning of the response-locked time window). Importantly, the polarity of these two ERP effects (i.e., 400 to 500 ms post-stimulus onset and -500 to -400 ms pre-response onset) were reversed (as evident by the signs of the t values shown in the corresponding tables below) which was the same as the waveform patterns previously observed. These additional results suggest that the variation in participant numbers (e.g., N = 28 in stimulus-locked analysis vs. N = 26 in response-locked analysis) might probably not be a crucial factor contributing to the observed contrastive waveform patterns. 

####For tables of Additional Analyses 1-3, please refer to the 'Response to Reviewers' file, named "response letter 20231109.docx". ####

Nevertheless, a further screening at the trial level by retaining only those trials that concurrently fulfilled the inclusion criterion (e.g., < ±120µV) of both stimulus-locked and response-locked analyses has not been performed. We refrained from performing this as to prevent the issue of double screening which may further reduce the data size and unavoidably limit our analyses to a restricted set of data sample. We do admit that it might be a limitation which requires further study. Future research is needed to further examine the different approaches to performing stimulus-locked and response-locked EEG analyses. Nevertheless, as the focus of this study is the relatively time course of syllabic and tonal encoding in Cantonese spoken word production, the results currently reported should be clear enough to inform this target issue. 

Reviewer #2 (Cont.): 

Second, the stimulus selection/control in the current experiment was actually not very perfect. I noticed that some of the target items contained two different syllables with the same tone (e.g., fei1 kei1, 'plane'). Since the tone is the focus of the current study, it will be better to try to avoid such items in order to obtain a more reliable effect, as it is unclear whether the tone itself could potentially show some effect in ERP. Even though previous studies did not report the tonal overlap alone could produce priming effect in behavioural data (in Cantonese and Mandarin), there is some evidence that the tonal overlap alone may potentially cause an interference effect in Mandarin (see Chen et al., 2002). Moreover, the rime sharing (e.g., fei1 kei1, 'plane') may also introduce some unknown influence to the processing.

Response:

Once again, we thank Reviewer 2 for this valuable comment. It is indeed the case that for some of the targets, their two syllables share the same tone (or having a similar sub-syllabic component). This is also a constraint that we faced when constructing the materials for this study. Given there are only six lexical tones in Cantonese, it is not easy (if not impossible) to find good enough of disyllabic targets, composed by two syllables having different tones, which are familiar to the participants (based on the rating results) and can easily be picturized and named. In addition, it also requires to have a matched atonal syllable available (a similar syllable but with a different tone) to form the atonal syllable distractor condition. Given the bulk of evidence in the literature suggests that the atonal syllable is the primary planning unit in Cantonese and Mandarin Chinese speech production, we therefore matched the two syllables of each of our disyllabic targets at the atonal syllable level (i.e., each target must be composed by two different atonal syllables). Furthermore, as a within-items design was adopted in this study, that is, the same set of targets were used across different distractor type conditions. So, if there were indeed effects from “within” the targets, such effects should be constant across different conditions. However, one may concern if these potential within-target effects would interact with the priming effects in different ways across different conditions. This is an empirical question requires further investigation and we admit that further research is needed to examine this possibility. We thank Reviewer 2 for this comment. A footnote has therefore been added in the revised manuscript in response to this comment.

---

## [Editor Report · Decision Letter 2]

20 Nov 2023

Tonal and syllabic encoding in overt Cantonese Chinese speech production: An ERP study

PONE-D-23-08190R2

Dear Dr. Chiu,

We’re pleased to inform you that your manuscript has been judged scientifically suitable for publication and will be formally accepted for publication once it meets all outstanding technical requirements.

Kind regards,

Caicai Zhang

Academic Editor

PLOS ONE
---

## [Editor Report · Acceptance letter]

4 Dec 2023

PONE-D-23-08190R2 

Tonal and syllabic encoding in overt Cantonese Chinese speech production: An ERP study 

Dear Dr. Chiu:

I'm pleased to inform you that your manuscript has been deemed suitable for publication in PLOS ONE. Congratulations! Your manuscript is now with our production department. 

Kind regards, 

on behalf of

Dr. Caicai Zhang 

Academic Editor

PLOS ONE